# Blockade of *PGK1* and *ALDOA* enhances bilirubin control of Th17 cells in Crohn's disease

Marta Vuerich[1,7], Na Wang [1,2,3,7], Jonathon J. Graham[1,7], Li Gao[1,4,7], Wei Zhang[1,4,7], Ahmadreza Kalbasi[1], Lina Zhang[1], Eva Csizmadia[1], Jason Hristopoulos[5], Yun Ma[6], Efi Kokkotou[5], Adam S. Cheifetz[5], Simon C. Robson [1,5] & Maria Serena Longhi [1✉]

Unconjugated bilirubin (UCB) confers Th17-cells immunosuppressive features by activating aryl-hydrocarbon-receptor, a modulator of toxin and adaptive immune responses. In Crohn's disease, Th17-cells fail to acquire regulatory properties in response to UCB, remaining at an inflammatory/pathogenic state. Here we show that UCB modulates Th17-cell metabolism by limiting glycolysis and through downregulation of glycolysis-related genes, namely phosphoglycerate-kinase-1 (*PGK1*) and aldolase-A (*ALDOA*). Th17-cells of Crohn's disease patients display heightened *PGK1* and *ALDOA* and defective response to UCB. Silencing of *PGK1* or *ALDOA* restores Th17-cell response to UCB, as reflected by increase in immunoregulatory markers like FOXP3, IL-10 and CD39. In vivo, *PGK1* and *ALDOA* silencing enhances UCB salutary effects in trinitro-benzene-sulfonic-acid-induced colitis in *NOD/scid/gamma* humanized mice where control over disease activity and enhanced immunoregulatory phenotypes are achieved. *PGK1* and/or *ALDOA* blockade might have therapeutic effects in Crohn's disease by favoring acquisition of regulatory properties by Th17-cells along with control over their pathogenic potential.

[1] Department of Anesthesia, Critical Care & Pain Medicine, Beth Israel Deaconess Medical Center, Harvard Medical School, Boston, MA, USA. [2] Department of Hematology, Shandong Provincial Hospital Affiliated to Shandong First Medical University, Shandong, China. [3] School of Medicine, Shandong University, Jinan, Shandong, China. [4] Department of Endocrinology and Metabolism, The First Affiliated Hospital of Shandong First Medical University & Shandong Provincial Qianfoshan Hospital, Jinan, Shandong, China. [5] Department of Medicine, Division of Gastroenterology, Beth Israel Deaconess Medical Center, Harvard Medical School, Boston, MA, USA. [6] Institute of Liver Studies, School of Immunology & Microbial Sciences, Faculty of Life Sciences & Medicine, King's College London, King's College Hospital, London, UK. [7] These authors contributed equally: Marta Vuerich, Na Wang, Jonathon J. Graham, Li Gao, Wei Zhang. ✉email: mlonghi@bidmc.harvard.edu

Unconjugated bilirubin (UCB) is a tetrapyrrole compound, which derives from the catabolism of heme. Heme oxygenase-1 catalyzes the degradation of heme to ferrous iron, carbon monoxide, and biliverdin, the latter converted to UCB by biliverdin reductase. When present at physiologic concentrations, UCB displays antioxidant and immunoregulatory effects[1–4]. We have previously shown that administration of UCB ameliorates the course of dextran sulfate-sodium (DSS)-induced colitis during the recovery phase of the disease[5]. Additional studies have shown that hyaluronic acid-bilirubin nanoparticles shape the gut microbiome by favoring bacterial species like *Akkermansia Muciniphila*, *Clostridium XIVa*, and lactobacilli[6], which are associated with protection from inflammatory bowel disease. Intravenous injection of bilirubin-based nanoparticles results in beneficial effects in mice with DSS colitis, further supporting the immunosuppressive properties of UCB in the control of acute colon inflammation[7]. Additional evidence of the immunoregulatory effects of bilirubin has been provided by clinical studies, showing decreased levels of this immunometabolite in the serum of inflammatory bowel disease patients[8–12]. Protection from developing or delay in manifesting Crohn's disease has been reported in subjects homozygous for the UDP-glucuronosyl transferase*28 allele, which is associated with high bilirubin levels[12,13].

UCB immunoregulatory effects depend, at least in part, on the activation of aryl hydrocarbon receptor (AhR), for which UCB serves as an endogenous ligand[5,14]. AhR is a mediator of toxin responses and a modulator of adaptive immunity, including Tregs and Th17 cells[15,16]. Upon activation through exogenous and endogenous ligands, like 6-formylindolo[3,2b]carbazole[16], 2,3,7,8-tetracholorodibenzo-p-dioxin[15], tryptophan metabolites and UCB[5], AhR controls a variety of biological processes, including regulation of xenobiotic metabolizing enzymes, the ATP-binding cassette family of drug transporters and cytokines like IL-10, IL-17, and IL-22[17].

Our previous studies revealed that Th17 cells obtained from patients with Crohn's disease are poorly responsive to AhR activation by UCB, as demonstrated by failure to undergo regulation and acquire immunosuppressive properties[5,18].

Besides the mechanisms mentioned above, there is evidence that activation of AhR may result in the modulation of cell metabolism[19,20]. Norisoboldine, a natural AhR agonist, promotes regulatory T cell (Treg) differentiation and disease remission in DSS colitis by repressing glycolysis and decreasing the levels of hexokinase II and glucose transporter 1[21].

Based on those earlier reports, we hypothesized that UCB exerts immunoregulatory effects in Th17 cells by modulating their metabolism. We also sought to investigate whether the impaired response of Th17 cells to UCB resulting in metabolic alterations plays a role in the pathogenesis of Crohn's disease.

Here we report that UCB controls glycolysis and downregulates glycolysis-related genes, mainly phosphoglycerate-kinase-1 (*PGK1*) and aldolase A (*ALDOA*), these effects being apparent in Th17 cells obtained from healthy individuals but not in those derived from Crohn's disease patients. Silencing of *PGK1* and *ALDOA* restores Th17 cell response to UCB in vitro, as reflected by an increase in the expression of immunoregulatory markers like the ectoenzyme CD39; and boosts UCB immunosuppressive properties in vivo, in an experimental model of colitis in humanized mice.

## Results

### UCB curbs glycolysis in Th17 cells of healthy subjects. We have previously shown that UCB serves as AhR endogenous ligand and confers immunosuppressive properties to Th17 cells obtained

from healthy controls, whereas these properties are impaired in Th17 cells derived from the peripheral blood and lamina propria of Crohn's disease patients[5,18]. To dissect the mechanisms underlying this impairment, we tested whether UCB might have an impact on the gene metabolic profile of Th17 cells. Th17 cells were differentiated from CD4 cells obtained from the peripheral blood of healthy controls and Crohn's disease patients, upon in vitro exposure to IL-6, IL-1β, and TGF-β[5,18,22].

After polarization, the frequency of Th17 cells ranged from 10–15% in both healthy subjects and Crohn's disease patients (Supplementary Fig. 1a, b), as previously reported by us and others[5,23].

Initial analysis of 180 metabolic genes in Th17 cells at baseline, prior to UCB exposure, identified 19 differentially expressed genes (DEGs) between healthy controls and patients with Crohn's disease (Fig. 1a). Of all the gene pathways tested, Kegg glycolysis appeared to be the most markedly altered in Th17 cells of Crohn's disease patients, when compared to healthy controls (Fig. 1b, c). Additional metabolic pathways that underwent significant changes in Crohn's disease included the mammalian target of rapamycin (mTOR) and glucose metabolism (Fig. 1b). Exposure of Th17 cells to UCB for the last 6 h of culture resulted in metabolic changes in both healthy controls and patients (Fig. 1d). These changes mainly impacted genes associated with Kegg glycolysis (Fig. 1e and Supplementary Fig. 2a). UCB addition also resulted in a significant decrease in extracellular acidification rate (ECAR) and oxygen consumption rate (OCR) in Th17 cells from healthy subjects (Fig. 2a). When considering Th17 cells derived from Crohn's disease patients, no significant changes in ECAR and OCR were noted upon exposure to UCB (Fig. 2a). There were no significant differences in ECAR and OCR between untreated Th17 cells of healthy subjects and Crohn's disease patients at any of the time points tested ($P > 0.05$ in all cases). UCB decreased the lactate levels in culture supernatants of Th17 cells obtained from healthy controls (Fig. 2b), while no significant differences were noted in the levels of lactate in supernatants of Th17 cells obtained from patients with Crohn's disease (Fig. 2b), indicating impaired Th17 cell response to UCB. The lactate levels in the supernatant of untreated Th17 cells from controls and patients were similar ($12.5 \pm 1.2$ vs $12.7 \pm 1.2$, $P = 0.8$). There were no differences in the levels of solute carrier family 2 member 1 (*SLC2A1*) (encoding for the glucose transporter protein type 1) between untreated and UCB treated cells of healthy subjects and Crohn's disease patients (Supplementary Fig. 2b). UCB, however, reduced the glucose uptake in Th17 cells of healthy subjects, while not having any impact on Th17 cells obtained from Crohn's disease patients (Supplementary Fig. 2c).

Among the Kegg glycolysis DEGs, *PGK1* and *ALDOA* were the genes most significantly impacted by exposure to UCB in both controls and Crohn's disease patients (Fig. 1e and Supplementary Fig. 2a).

At baseline, prior to UCB addition, *PGK1* and *ALDOA* mRNA levels were higher in peripheral blood derived Th17 cells of Crohn's disease patients, when compared to controls (Fig. 3a). When analyzing Th17 cells derived from the lamina propria, *ALDOA* levels were significantly increased in Th17 cells obtained from inflamed, as compared to non-inflamed biopsied areas (Fig. 3b); whereas levels of *PGK1* were similar in the two biopsied regions (Fig. 3b). In Crohn's disease, there was a positive correlation between the levels of *PGK1* or *ALDOA* and Montreal type (Fig. 3c).

The addition of UCB resulted in a marked decrease in *PGK1* and *ALDOA* mRNA levels in peripheral blood derived Th17 cells obtained from healthy controls (Fig. 3d and Supplementary Fig. 3a). Decrease in *PGK1* and *ALDOA* mRNA levels was also noted in Th17 cells of Crohn's disease patients although to a

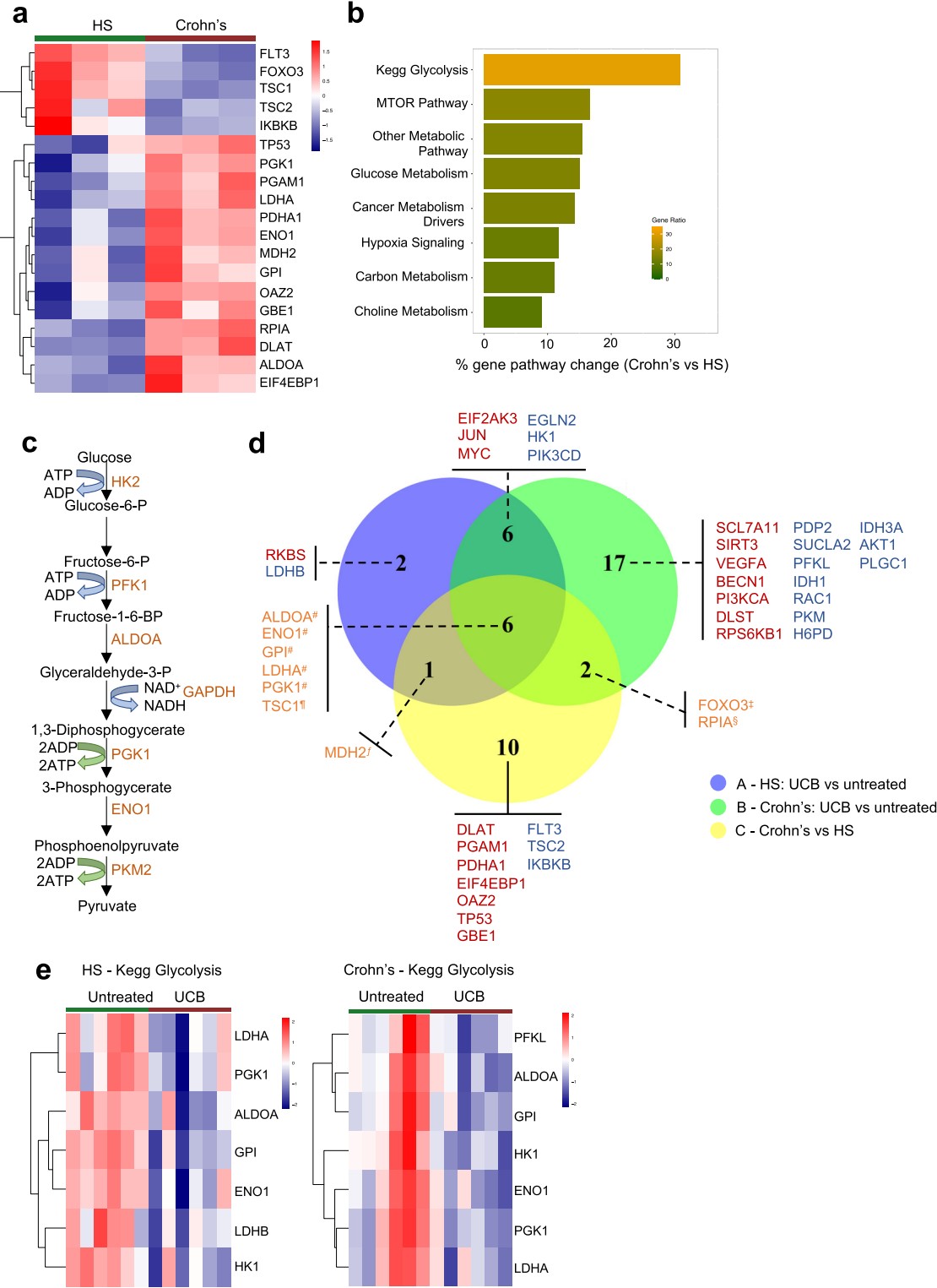

lesser extent than in healthy controls (Fig. 3d and Supplementary Fig. 3a). Exposure to UCB decreased PGK1 and ALDOA mean fluorescence intensity (MFI) in Th17 cells of healthy subjects (Supplementary Fig. 4a, b) and tended to decrease ALDOA MFI in Th17 cells of patients (Supplementary Fig. 4b). When analyzing Th17 cells obtained from patients' lamina propria, decrease in both *PGK1* and *ALDOA* mRNA levels was noted in cells obtained from non-inflamed biopsied areas, following exposure to UCB (Fig. 3e and Supplementary Fig. 3b).

Importantly, exposure to UCB could reduce *PGK1* and *ALDOA* levels in Th17 cells derived from the peripheral blood of healthy controls, after challenge with high concentration glucose; whereas such an effect was not present when considering Th17 cells derived from Crohn's disease patients (Fig. 3f and Supplementary Fig. 3c, d).

The effects of UCB on *Pgk1* and *Aldoa* were also tested in mice with DSS-induced experimental colitis. Daily administration of UCB decreased the disease activity index during recovery, when

**Fig. 1 Unconjugated bilirubin curbs glycolysis-related genes in Th17 cells from healthy controls.** Th17 cells, obtained upon the polarization of peripheral blood CD4 cells, were exposed to unconjugated bilirubin (UCB) for 6 h and then harvested for metabolic gene profiling using NanoString. **a** Heatmap showing differentially expressed genes (DEGs) included in the nCounter Vantage RNA Cancer Metabolism panel in Th17 cells obtained from the peripheral blood of healthy subjects (HS, $n = 3$) and patients with Crohn's disease ($n = 3$). Upregulated and downregulated genes are shown in red and blue, respectively. DEGs were defined based on $P$ value $\leq 0.05$ (false discovery rate $\leq 0.05$ in all cases). **b** Gene pathway ranking in Th17 cells was based on the percentage of DEGs in each pathway. Pathways where $\geq 50\%$ of genes displayed changes (i.e., upregulation or downregulation), for which the $P$ value was $\leq 0.05$ when compared to HS samples, are indicated in a brighter color. **c** Kegg glycolysis schematic. **d** Venn diagram showing DEGs when comparing UCB treated and untreated Th17 cells in HS (A - purple) and Crohn's disease patients (B - green) and when comparing Crohn's disease patients and HS (C - yellow). Red: genes upregulated in both groups; blue: genes downregulated in both groups; # decrease in "A" and "B", increase in "C"; ¶ increase in "A" and "B", decrease in "C"; f decrease in "A", increase in "C"; ‡ increase in "B", decrease in "C"; § decrease in "B", increase in "C". **e** Heatmaps showing DEGs within the Kegg glycolysis pathway in untreated and UCB treated Th17 cells in HS ($n = 6$) and Crohn's disease patients ($n = 6$). Upregulated and downregulated genes are shown in red and blue, respectively. UCB downregulates glycolysis-associated genes and this effect is more evident in HS than in Crohn's disease. All samples subjected to NanoString analysis were run in triplicate and experiments were performed independently three times.

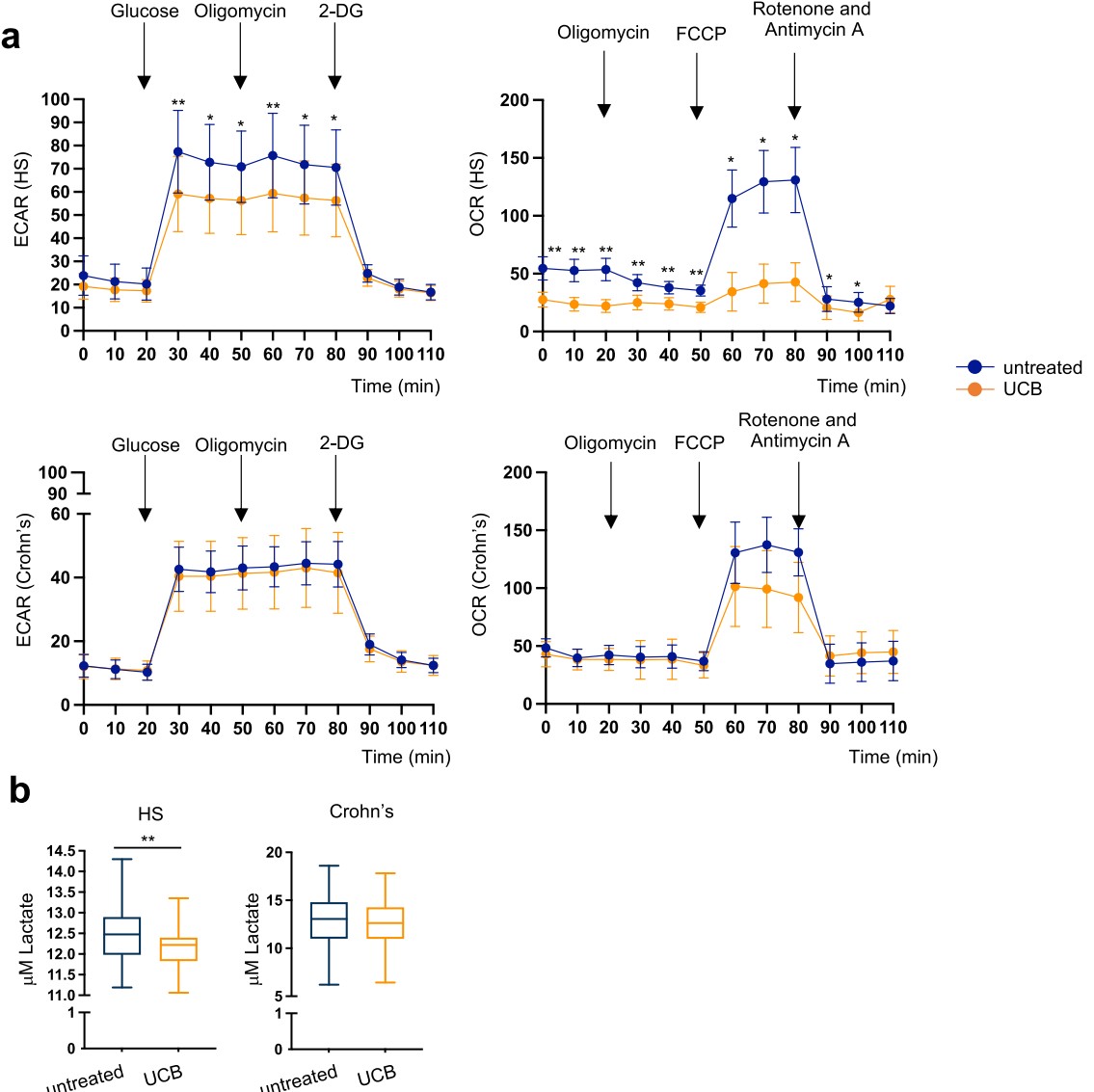

**Fig. 2 Unconjugated bilirubin limits Th17 cell glycolysis rate.** Extracellular acidification rate (ECAR) and oxygen consumption rate (OCR) of Th17 cells in the absence or presence of UCB was determined by Seahorse. **a** Mean ± SEM ECAR and OCR measured by Seahorse in untreated and UCB treated Th17 cells of HS ($n = 9$ for ECAR and $n = 10$ for OCR) and Crohn's disease patients ($n = 7$). UCB effectively reduces ECAR and OCR in Th17 cells of HS, this effect being less marked in Th17 cells obtained from patients with Crohn's disease. **b** Box-and-whisker plots representing lactate levels in the culture supernatants of untreated and UCB treated Th17 cells from HS ($n = 10$) and patients with Crohn's disease ($n = 10$). *$P \leq 0.05$ and **$P \leq 0.01$ using two-sided unpaired $t$-test.

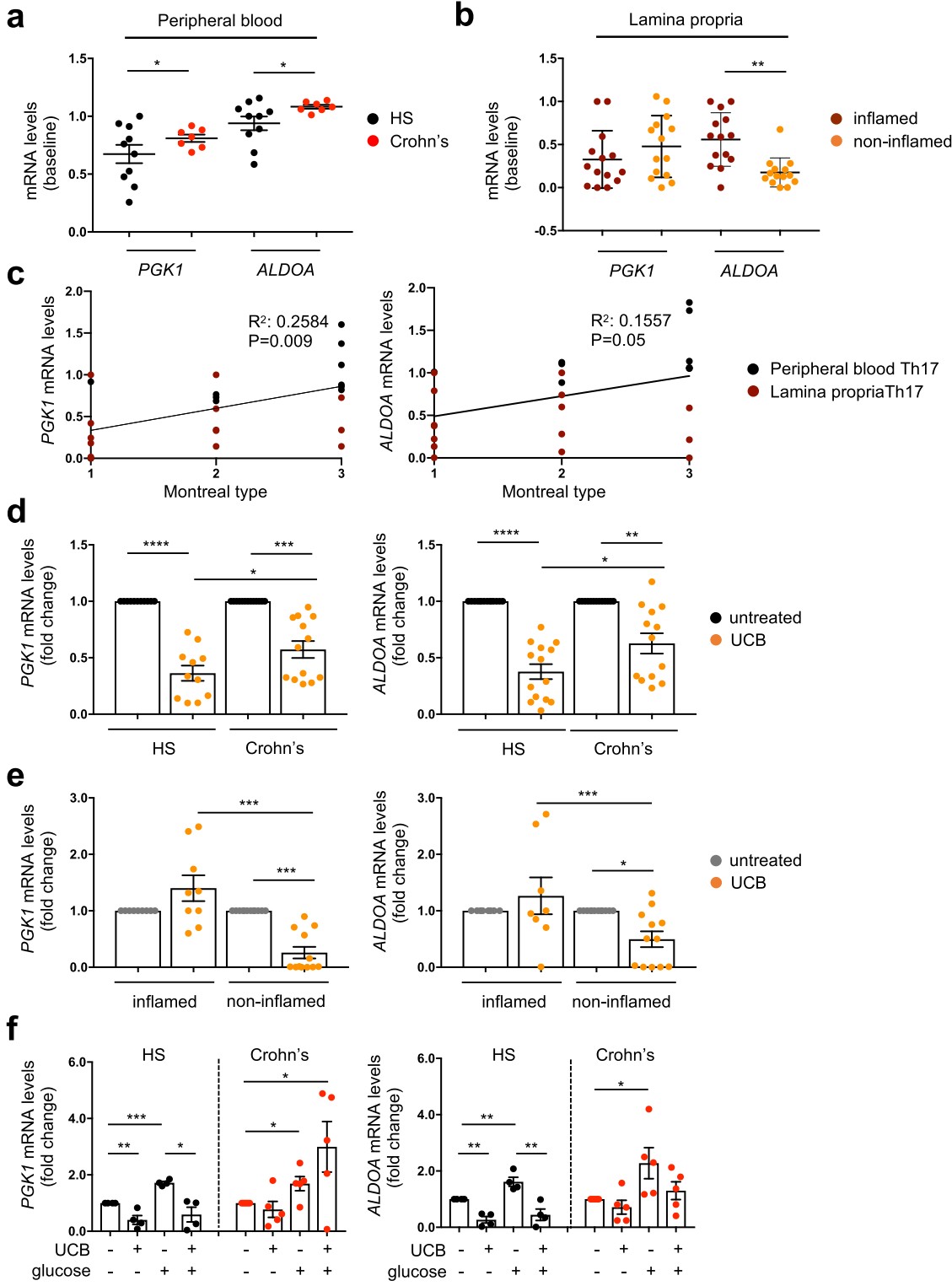

DSS was replaced with standard drinking water (Fig. 4a, b). UCB administration also resulted in greater colon length (Fig. 4c), lower histology score (Fig. 4d), and reduced levels of pro-inflammatory cytokines in intra-epithelial (IEL) and LP derived CD4 cells at the harvest (Supplementary Fig. 5a–d). Decrease in *Pgk1* and *Aldoa* mRNA levels was noted in CD4 cells obtained from the mesenteric lymph nodes (MLNs) and LP lymphocytes of UCB treated mice (Fig. 4e).

In both healthy controls and Crohn's disease patients, exposure of Th17 cells to UCB left unchanged the levels of

peroxisome proliferator-activated receptors α (*PPARα*) and γ (*PPARγ*) (Supplementary Fig. 6a, b) that were previously linked to control of glycolysis[24] and inhibition of glycolytic genes like *PGK1*[25]. No changes in the expression of adenosine monophosphate (AMP)-activated protein kinase (*AMPK*) or *mTOR*, which modulate intestinal inflammation through the regulation of metabolism[26–30], were noted in Th17 cells of controls and patients upon exposure to UCB (Supplementary Fig. 6c, d).

Overall, these data indicate that UCB modulates Th17 cell metabolic status, mainly by impacting glycolysis and through the

**Fig. 3 UCB downregulates *PGK1* and *ALDOA* in Th17 cells from healthy subjects.** Mean ± SEM *PGK1* and *ALDOA* mRNA levels in untreated Th17 cells were obtained from (**a**) the peripheral blood of healthy subjects ($n = 10$) and Crohn's disease patients ($n = 7$) and from (**b**) inflamed and non-inflamed biopsied areas of Crohn's disease patients ($n = 14$). Baseline levels of *PGK1* and *ALDOA* are heightened in Th17 cells of Crohn's disease patients. Levels of *ALDOA* in Th17 cells derived from non-inflamed biopsied areas are lower than in Th17 cells obtained from inflamed biopsied areas. **c** Correlation between *PGK1* or *ALDOA* mRNA levels with Montreal type (correlation made using Pearson correlation coefficient). **d** Mean ± SEM *PGK1* and *ALDOA* mRNA levels in untreated and unconjugated bilirubin (UCB) treated Th17 cells obtained from the peripheral blood of healthy subjects ($n = 11$ for *PGK1* and $n = 13$ for *ALDOA*) and Crohn's disease patients ($n = 13$ for *PGK1* and $n = 12$ for *ALDOA*) and from (**e**) inflamed ($n = 9$ for *PGK1* and $n = 8$ for *ALDOA*) and non-inflamed ($n = 12$) biopsied areas of Crohn's disease patients. **f** Mean ± SEM *PGK1* and *ALDOA* mRNA levels in Th17 cells, untreated or exposed to UCB, glucose, and UCB plus glucose ($n = 4$ HS and $n = 5$ Crohn's disease patients). In '**d**–**f**', results are expressed as fold change compared to untreated Th17 cells. *$P \leq 0.05$, **$P \leq 0.01$, ***$P \leq 0.001$, and ****$P \leq 0.0001$ using two-sided unpaired $t$-test (**a**, **b**) and one-way ANOVA test, followed by Tukey's multiple comparison test (**d**–**f**).

downregulation of glycolysis-associated genes like *PGK1* and *ALDOA*. These changes are prominent in cells obtained from healthy individuals but not in those obtained from Crohn's disease patients.

**Silencing of *PGK1* or *ALDOA* boosts Th17 cell immunoregulatory properties in Crohn's disease.** To determine the extent to which altered glucose utilization modulates Th17 cell response to UCB, we exposed peripheral blood derived Th17 cells to 2-deoxy-2-fluoro-beta-D-arabinonucleic acid (FANA) oligonucleotides to specifically silence *PGK1* or *ALDOA* (hereafter FANA-*PGK1* and FANA-*ALDOA*), in the absence or presence of UCB. We have previously shown that UCB boosts the levels of FOXP3, IL-10, and CD39, an ectoenzyme that hydrolyzes ATP/ADP, ultimately generating immunosuppressive adenosine[5,18]. Therefore, these markers were used as readouts to assess the response of Th17 cells to UCB, alone or in combination with FANA oligonucleotides. Compared to scramble, the addition of UCB and FANA-*PGK1* or FANA-*ALDOA* boosted the levels of *ENTPD1* (encoding for CD39), *FOXP3*, and *IL10* in both controls and patients (Fig. 5a–c), whereas the addition of UCB alone increased *ENTPD1*, *FOXP3* and *IL10* expression in controls (Fig. 5a–c) and boosted *FOXP3* levels in Th17 cells derived from Crohn's disease patients (Fig. 5b). In Crohn's disease, levels of *ENTPD1* and *IL10* were higher in Th17 cells exposed to UCB and FANA-*PGK1* or FANA-*ALDOA* than in those exposed to UCB alone; this increase was significant in the case of *IL10* and trended towards significance for *ENTPD1* (Fig. 5a, c). Compared to scramble but not UCB, the addition of FANA-*PGK1* or FANA-*ALDOA* alone increased *ENTPD1* in healthy control and *IL10* in healthy control and patient samples (Fig. 5a, c). No differences were noted in the levels of *IL17A* upon the addition of FANA oligonucleotides in the absence or presence of UCB (Supplementary Fig. 7a).

No significant changes in *ENTPD1* mRNA levels were noted in Tregs, Th1, and Th2 cells upon exposure to UCB in combination with FANA-*PGK1* or FANA-*ALDOA* (Supplementary Fig. 7b, d, f). In Tregs, the addition of FANA oligonucleotides in the absence or presence of UCB resulted in levels of *FOXP3* that were comparable to those obtained in the presence of UCB only (Supplementary Fig. 7c). No significant changes were noted in the levels of IFNγ and IL-4 in Th1 and Th2 cells respectively upon exposure to UCB and FANA oligonucleotides, alone or in combination (Supplementary Fig. 7e, g).

Analysis of cell phenotypes by flow cytometry showed that the frequency of Th17 cells (Supplementary Fig. 1a, b) positive for CD39, FOXP3, or IL-10 increased upon exposure to UCB and FANA-*PGK1* or FANA-*ALDOA*, when compared with cells exposed to scramble (Fig. 5d–f and Supplementary Fig. 8a–c). In both controls and patients, this increase was statistically significant when considering CD39+ or FOXP3+ cells (Fig. 5d, e), while trended towards significance in the case of IL-10 producing cells within polarized Th17 cells (Fig. 5f). Exposure to UCB alone

significantly increased the frequency of CD39+ cells and that of FOXP3+ and IL-10+ cells among Th17 lymphocytes in controls but not in patients with Crohn's disease (Fig. 5d–f and Supplementary Fig. 8a–c). Compared to scramble, but not UCB, the addition of FANA oligonucleotides increased the frequency of CD39+ cells in controls (Fig. 5d and Supplementary Fig. 8a) and the proportion of IL-10 producing cells within polarized Th17 cells in patients (Fig. 5f and Supplementary Fig. 8c).

To determine whether silencing of *PGK1* and/or *ALDOA* impact the differentiation of CD4 lymphocytes into Th17 cells, FANA-*PGK1* and FANA-*ALDOA* were added to peripheral blood derived CD4 cells for the last 72 h of their polarization into Th17 cells. The addition of FANA-*PGK1* or FANA-*ALDOA* reduced the frequency of RORC+ cells within CD4 lymphocytes obtained from patients with Crohn's disease, this decrease trending towards significance (Supplementary Fig. 9a). Such an effect was not noted upon the addition of FANA oligonucleotides to CD4 cells obtained from healthy controls (Supplementary Fig. 9a). There were no differences in the frequency of FOXP3+ lymphocytes after addition of FANA oligonucleotides to CD4 cells obtained from healthy controls and patients with Crohn's disease (Supplementary Fig. 9b).

Overall, these data indicate that blockade of *PGK1* or *ALDOA* using FANA oligonucleotides restores the response of Th17 cells derived from Crohn's disease patients to UCB; this is reflected by higher levels of CD39, FOXP3, and IL-10, increased proportions of CD39+ cells and heightened frequencies of FOXP3+ and IL-10+ lymphocytes within polarized Th17 cells.

**Silencing of *PGK1* or *ALDOA* enhances UCB immunoregulatory effects in experimental colitis in humanized mice.** To verify the impact of these findings in vivo, we administered FANA-*PGK1* or FANA-*ALDOA*, alone or in combination with UCB, into *NOD/scid/gamma* mice, preemptively reconstituted with CD4 cells from one healthy blood donor and subsequently exposed to trinitrobenzene sulfonic acid (TNBS) (Supplementary Fig. 10a), as we described previously[22]. Levels of *PGK1* and *ALDOA* in CD4 cells were measured by qRT-PCR prior to injection. Compared to mice treated with scramble, mice receiving a combination of UCB and FANA-*PGK1* or FANA-*ALDOA* had a better disease course, as reflected by lower disease activity index (Fig. 6a), increased colon length (Fig. 6b), decreased histology score, and CD3 cell infiltration (Fig. 6c, d) at the harvest. No significant changes were noted in the disease activity index and histology score of mice administered FANA-*PGK1* or FANA-*ALDOA* alone (Fig. 6a, c) when compared to scramble treated mice.

Analysis of immune cell compartments in different organs showed that mice exposed to a combination of FANA oligonucleotides and UCB had decreased frequencies of IL-17A+ cells in MLN (Fig. 6e and Supplementary Fig. 11a), higher proportions of FOXP3+ lymphocytes in the spleen (Fig. 6f and Supplementary Fig. 11b), reduced proportion of IFNγ producing lymphocytes in MLNs and IELs (Supplementary Figs. 10b, 11c) and

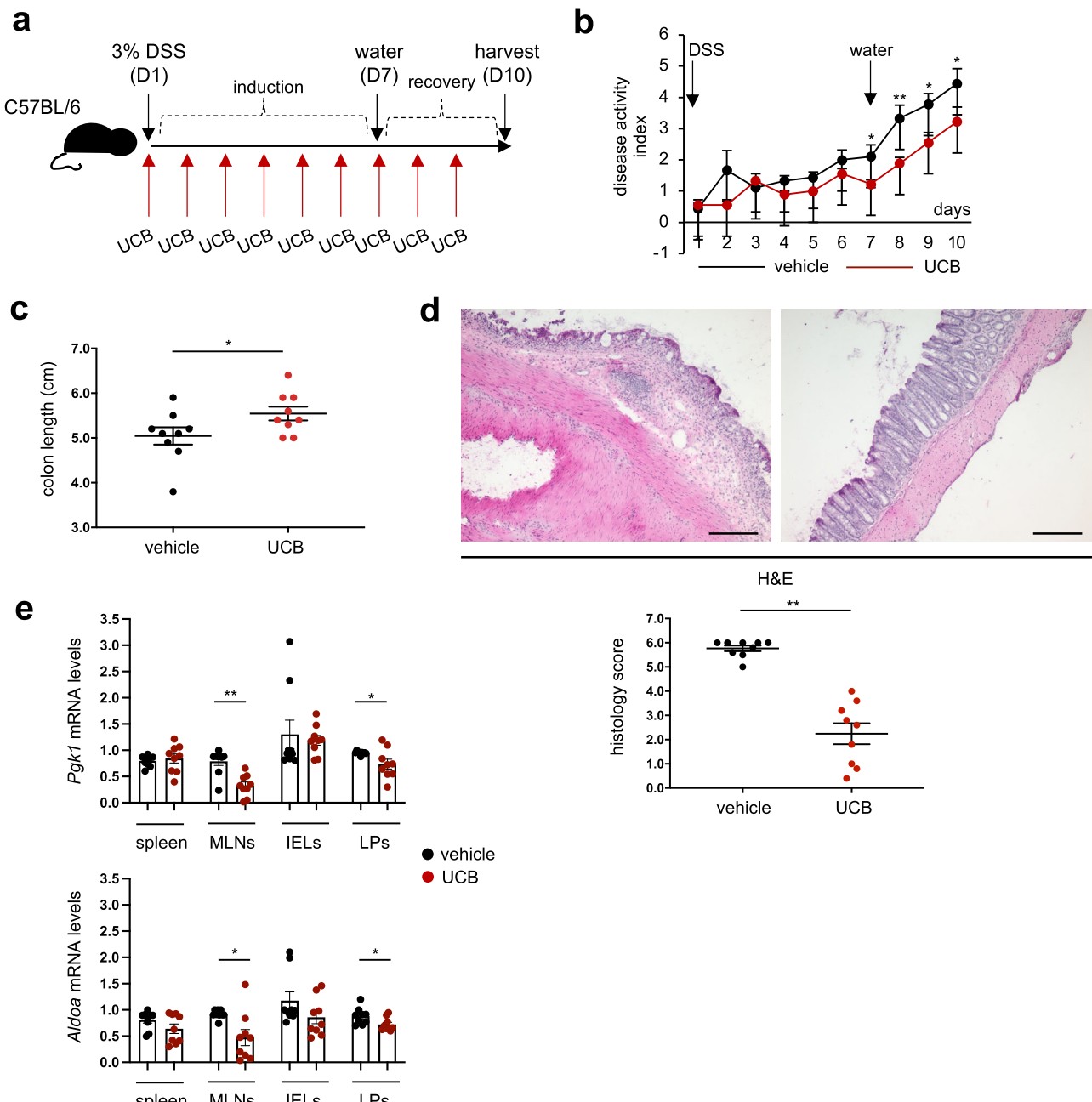

**Fig. 4 UCB decreases *Pgk1* and *Aldoa* levels in CD4 cells of mice with DSS colitis. a** Wild-type mice were exposed to 3% dextran sulfate sodium (DSS) for 6 days. DSS was then replaced with normal drinking water for an additional 4 days. For the whole experiment duration, mice were administered either vehicle ($n = 9$) or unconjugated bilirubin (UCB) ($n = 9$) at 20 μmol/kg/day. **b** Mean ± SEM disease activity index in vehicle and UCB treated animals. **c** Mean ± SEM colon length (cm) at the time of harvesting. **d** Hematoxylin and Eosin (H&E) staining of colon sections from untreated and UCB treated mice (original magnification, ×10, scale bar: 200 μM); mean ± SEM histology score at the harvest is also shown. Mean ± SEM (**e**) *Pgk1* and *Aldoa* mRNA levels in CD4 lymphocytes isolated from the spleen, mesenteric lymph node (MLN), intra-epithelial (IEL), and lamina propria (LP). *$P \leq 0.05$ and **$P \leq 0.01$, using two-sided unpaired *t*-test.

lower frequencies of IL-10+ and total CD39+ cells in MLNs (Supplementary Fig. 10c, d and Supplementary Fig. 11d, e), when contrasted to scramble treated mice. When considering the CD39+ cell subfractions, we noted higher frequencies of IL-17A+ cells within the CD39+ compartment in the spleen (Fig. 6g and Supplementary Fig. 11f) and increased proportions of FOXP3+ cells within the CD39+ compartment in the spleen and MLNs of mice exposed to FANA oligonucleotides and UCB (Fig. 6h and Supplementary Fig. 11g). Compared to mice treated with UCB alone, mice receiving UCB and FANA oligonucleotides displayed

decreased IL-17A+ cells in LPs (Fig. 6e and Supplementary Fig. 11a) and higher FOXP3+ cells within the CD39+ subset in MLNs (Fig. 6h and Supplementary Fig. 11g). Treatment with FANA oligonucleotides only resulted in decreased proportions of FOXP3+ cells in IELs (Fig. 6e and Supplementary Fig. 11b), and increased frequencies of IFNγ+ cells in LP (Supplementary Figs. 10b, 11c), when compared with UCB or scramble treated animals.

Overall, these data indicate that silencing of *PGK1* and *ALDOA* has beneficial effects when combined with UCB in vivo, as reflected by ameliorated disease activity and increase in the

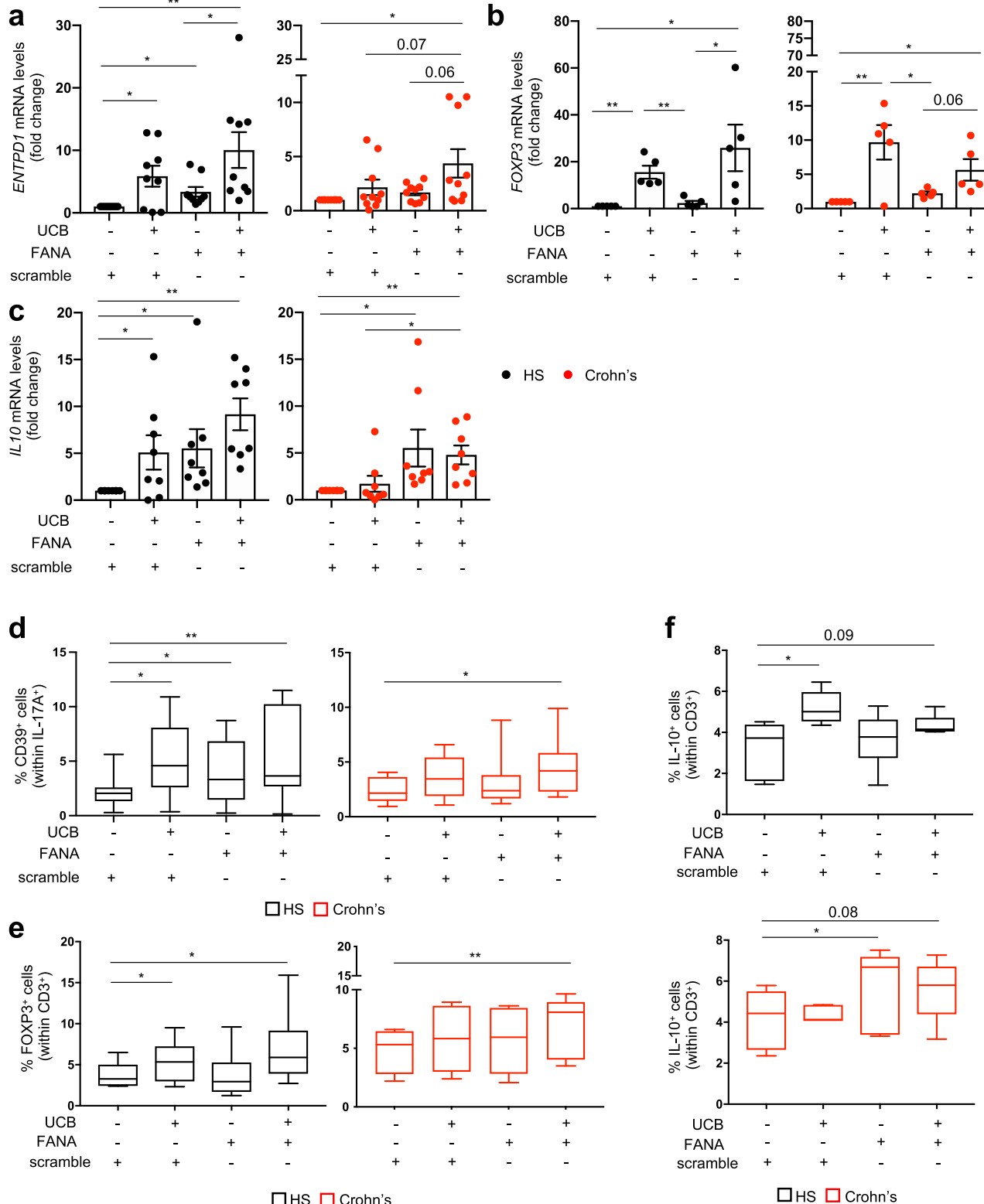

**Fig. 5 Silencing of *PGK1* and *ALDOA* boosts UCB immunoregulatory properties in Th17 cells derived from Crohn's disease patients.** Peripheral blood derived Th17 cells were exposed to FANA oligonucleotides, specifically silencing *PGK1* or *ALDOA*, alone or in combination with unconjugated bilirubin (UCB). Mean ± SEM '**a**' *ENTPD1* (encoding for CD39), '**b**' *FOXP3*, and '**c**' *IL10* mRNA levels in Th17 cells exposed to scramble, UCB, FANA oligonucleotides or FANA oligonucleotides plus UCB in healthy subjects (HS, $n = 9$ in **a**, $n = 5$ in **b**, and $n = 8$ in **c**) and Crohn's disease patients ($n = 10$ in **a**, $n = 5$ in **b**, and $n = 8$ in **c**). *$P \leq 0.05$ and **$P \leq 0.01$ using ANOVA repeated measures followed by Tukey's multiple comparison test. Results obtained in the presence of FANA-*PGK1* and FANA-*ALDOA* are pooled. Box-and-whisker plots representing the frequency of (**d**) CD39+ (% of IL-17A+), (**e**) FOXP3+ (% of CD3+), and (**f**) IL-10+ (% of CD3+) lymphocytes in polarized Th17 cells exposed to scramble, UCB, FANA oligonucleotides or FANA oligonucleotides plus UCB (HS, $n = 12$ in **d** and $n = 6$ in **e**, **f** Crohn's disease patients, $n = 10$ in **d** and $n = 5$ in **e**, **f**). Results obtained in the presence of FANA-*PGK1* or FANA-*ALDOA* are pooled. *$P \leq 0.05$ and **$P \leq 0.01$ using ANOVA repeated measures followed by Tukey's multiple comparison test.

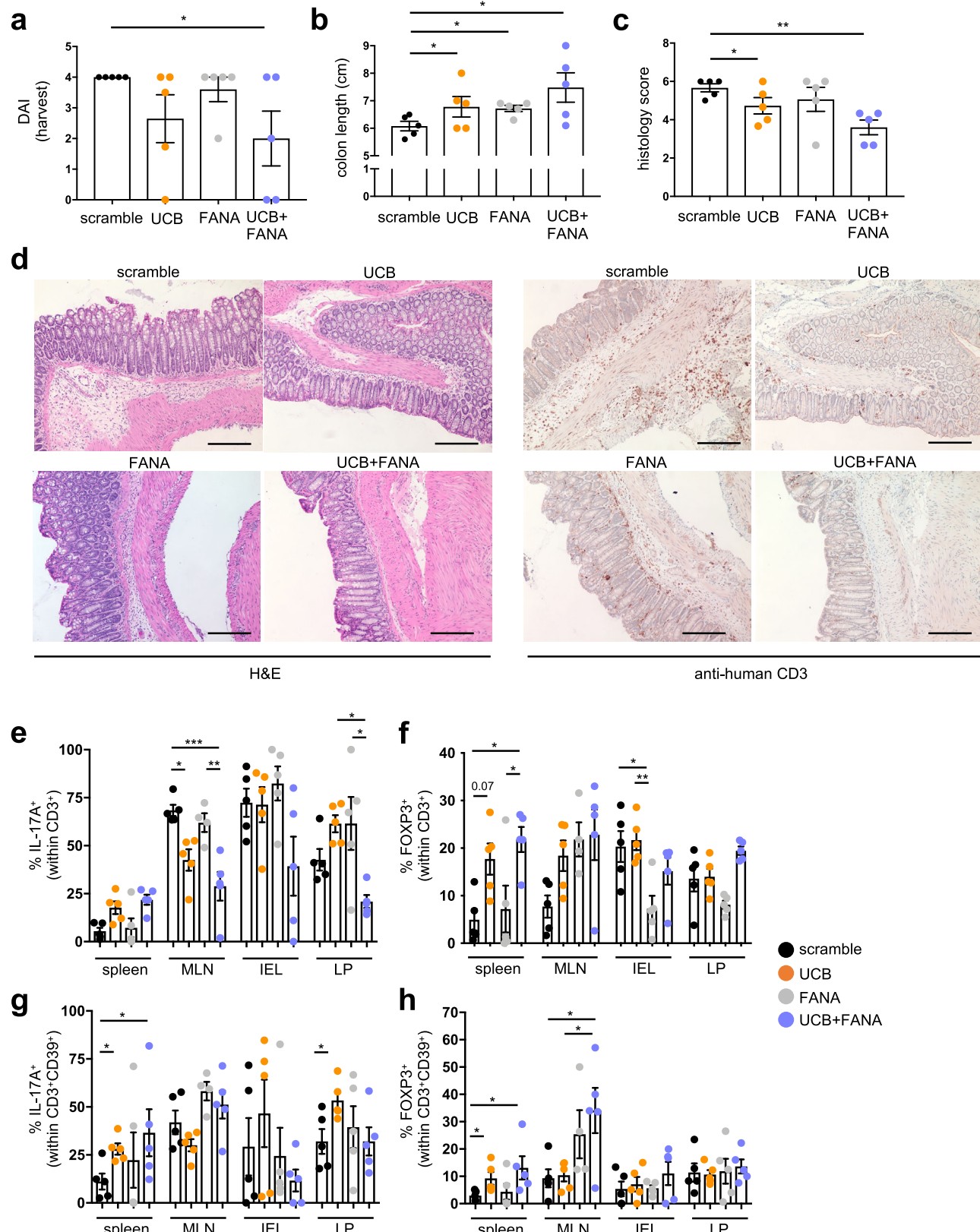

proportion of immunoregulatory subsets within CD4 cells of spleen and MLNs of *NOD/scid/gamma* humanized mice.

## Discussion

We provide evidence that UCB modulates the metabolism of Th17 cells by controlling glycolysis and favoring the acquisition of regulatory properties by this subset of effector lymphocytes. Control of glycolysis, which governs multiple biological processes culminating with the production of pro-inflammatory ATP, would result in attenuation of Th17 cell effector potential and pathogenicity. In healthy control derived Th17 cells, UCB control of glycolysis mainly occurs through downregulation of PGK1 and

**Fig. 6 Silencing of *PGK1* or *ALDOA* boosts UCB immunosuppressive effects in TNBS-induced experimental colitis in *NOD/scid/gamma* humanized mice.** *NOD/scid/gamma* (NSG) female recipients were injected with *PGK1*+ and *ALDOA*+ CD4 cells obtained from one healthy blood donor. After three weeks, mice were checked for human chimerism. Mice showing more than 10% human chimerism were sensitized to TNBS and one week later, administered a single enema of TNBS and a single intraperitoneal injection of a scramble, FANA-*PGK1* or FANA-*ALDOA*, unconjugated bilirubin (UCB) or a combination of UCB and FANA oligonucleotides. After 72 h, mice were sacrificed. **a** Mean ± SEM disease activity index at the time of harvest in TNBS mice treated with scramble (*n* = 5), UCB (*n* = 5), FANA oligonucleotides (*n* = 5), and FANA oligonucleotides in combination with UCB (*n* = 5). Mean ± SEM '**b**' colon length (cm) and '**c**' histology score at the harvest in scramble (*n* = 5), UCB (*n* = 5), FANA oligonucleotides (*n* = 5), and FANA oligonucleotides plus UCB treated (*n* = 5) mice. **d** Hematoxylin & Eosin (H&E) and anti-human CD3 staining of colon sections (original magnification, ×10, scale bar: 200 µM). Mean ± SEM frequency of '**e**' IL-17A+ and '**f**' FOXP3+ cells within the CD3+ cell compartment; and of '**g**' IL-17A+ and '**h**' FOXP3+ cells within the CD3+CD39+ lymphocyte subset in the spleen, mesenteric lymph node (MLN), intra-epithelial (IEL) and lamina propria (LP) lymphocytes at the time of harvest in scramble (*n* = 5), UCB (*n* = 5), FANA oligonucleotide (*n* = 5), or FANA oligonucleotide and UCB (*n* = 5) treated animals. Results obtained in the presence of FANA-*PGK1* or FANA-*ALDOA* are pooled. *$P \leq 0.05$, **$P \leq 0.01$, and ***$P \leq 0.001$ using one-way ANOVA followed by Tukey's multiple comparisons test.

ALDOA, decrease in ECAR at the functional level and reduced formation of lactate, the end-product of glycolysis. The decrease in glucose uptake observed in control Th17 cells upon exposure to UCB might also result from the inhibitory effects of this immunometabolite over glycolysis and related genes.

Importantly, control over PGK1 and ALDOA is also noted in the presence of high glucose concentrations, as in our challenge experiments. These effects are markedly attenuated, or even absent, in Th17 cells obtained from Crohn's disease patients, likely as a result of the inability of these cells to effectively respond to AhR activation[5,18,31]. The evidence that Th17 cells obtained from non-inflamed biopsied areas downregulate *PGK1* and *ALDOA* levels following UCB exposure postulates that the inflammatory cytokine milieu might be an additional important factor impacting Th17 cell ability to respond to AhR. Due to high levels of ATP in the milieu, Th17 cells fail to acquire a regulatory phenotype and therefore persist in an inflammatory state. In Crohn's disease, the lack of effective CD39 upregulation by Th17 cells following exposure to UCB further amplifies this inflammatory cascade, which is not adequately contrasted by adenosinergic signals initiated by CD39 mediated ATP hydrolysis.

The findings of decreased *Pgk1* and *Aldoa* mRNA levels in CD4 cells isolated from MLN and LP derived lymphocytes of UCB treated mice with DSS colitis further corroborate the evidence that UCB immunoregulatory properties depend, at least to some extent, on metabolic control and regulation of glycolysis. Future experiments should address the direct impact of UCB on Th17 cells in a T cell transfer model of colitis to evaluate whether also, in this context, UCB modulates inflammation and attenuates Th17 cell pathogenic potential through the control of glycolysis and glycolysis-related genes.

Previous reports have indicated that AhR controls glucose metabolism[19,20] and norisoboldine, a natural AhR agonist, was found to promote Treg differentiation and disease remission in DSS-induced colitis by repressing glycolysis[21]. Given that pathogenic Th17 cells rely on glycolysis for their energy requirements as other effector cell subsets[32,33], this might represent an important mechanism to control Th17 metabolism and, possibly, limiting the pathogenicity of this lymphocyte subpopulation. Importantly, our data indicate that UCB does not impact the expression of *PPARα* or *PPARγ* that could modulate inflammation through downregulation of glycolytic genes like *PGK1*[24,25,34]. Further, no changes in the levels of *AMPK* and *mTOR* were noted upon Th17 cell exposure to UCB, which was previously shown decreasing mTOR phosphorylation in the setting of cancer[35]. These findings suggest that, in our setting, downregulation of glycolytic genes likely results from AhR signaling activation. Future studies should determine whether *PPARα*, *PPARγ*, *AMPK*, and *mTOR* modulate glycolysis in Crohn's disease in an AhR-independent manner.

Silencing of *PGK1* and *ALDOA* ameliorates the response of Th17 cells to UCB. This is evident also in Th17 cells obtained from Crohn's disease patients, where higher levels of FOXP3 and IL-10 are noted upon exposure to UCB in association with FANA oligonucleotides targeting *PGK1* or *ALDOA*. Blockade of these genes boosts the expression of CD39 ectoenzyme, the levels of which are impaired in Th17 cells derived from the peripheral blood and lamina propria of Crohn's disease patients[36], as a result of altered regulation at the genetic and transcriptional levels[5,18,22,37]. In Crohn's disease, the altered response of Th17 cells to UCB also derives from high levels of HIF-1α that, in concert with a hypoxic environment, limits the responsiveness of these cells to AhR activation[18]. This results from the upregulation of drug transporters that favor the exit of immunometabolites like UCB out of cells[18] that, in turn, fail to upregulate CD39. It is plausible postulating that a high glycolysis rate might be the consequence of defective AhR control over metabolism, resulting from a hypoxic inflammatory environment and leading to high PGK1/ALDOA and decreased CD39 expression. Links between CD39 and purinergic mediators with metabolism have been reported before in the context of hepatocellular carcinoma, where *Cd39* deficiency was associated with increased production of lactate[38]; and in liver inflammation, where low CD39 and CD73 levels were noted in combination with upregulated hexokinase 2, pyruvate kinase M2 (PKM2) and lactate dehydrogenase[39]. Additional links were reported in the context of Treg adoptive transfer, where expanded Tregs were found to switch to aerobic glycolysis while enhancing their suppressor properties as a result of HIF-1α induced expression of CD73[40].

The effects of *PGK1* or *ALDOA* silencing on *ENTPD1* levels in the presence of UCB appeared to be specific to Th17 cells, as no increase in *ENTPD1* was noted in Th1 and Th2 cell subsets and in Tregs. The increase in the levels of FOXP3 upon exposure to FANA-*PGK1* and FANA-*ALDOA* suggests that, in Tregs, interference with glycolysis boosts a suppressive phenotype, which might be independent of AhR activation.

Silencing of *PGK1* and *ALDOA* has beneficial effects when combined with UCB in vivo, as reflected by ameliorated disease activity and an increase in the proportion of immunoregulatory subsets within CD4 cells of the systemic compartments. This approach not only ameliorates clinical parameters of colitis but also results in immunoregulatory changes of both effector and regulatory CD4 T cell subsets, as reflected by lower proportions of lymphocytes producing pro-inflammatory cytokines and an increase in the proportion of IL-17A+CD39+ and FOXP3+CD39+ cells in the spleen and MLNs. The lower frequency of IL-10+ and total CD39+ cells in MLNs of mice exposed to treatment with UCB and FANA-*PGK1* or FANA-*ALDOA* is likely compensated by a concomitant decrease in IFNγ+ and IL-17A+ lymphocytes and increased FOXP3+CD39+ cells in this compartment. Importantly, in Crohn's derived Th17 cells, silencing of *PGK1* or *ALDOA* tends to decrease the expression of RORC within CD4 cells during their differentiation, leaving unchanged the levels of FOXP3. This

suggests that blockade of these glycolysis-related genes exerts a more prominent control over the differentiation into Th17 cells rather than boosting polarization into Tregs.

Th17 cell differentiation and pathogenicity are linked to glycolysis and previous work has shown that deletion of the glycolytic enzyme PKM2 resulted in amelioration of autoimmune encephalomyelitis (EAE) by a decrease of Th17 cell-mediated inflammation[41]. In the same inflammatory setting, inhibition of CaMK4, which interacts directly with PKM2, was shown to modulate EAE[42]. Further, inhibition of glycolysis by targeting Gpi1, which was also upregulated in our setting (Fig. 1e and Supplementary Fig. 2A), could remove encephalitogenic and colitogenic Th17 cells[32]. Our data show that silencing of *PGK1* and *ALDOA* boosts UCB immunosuppressive properties in vivo and restores Th17 cell ability to respond to this immunometabolite in vitro. Failure of anti-IL-17 immunotherapy in autoimmune diseases, like Crohn's colitis[43–45], might also derive from a lack of effective control over glycolysis due to the inability of Th17 cells to adequately respond to UCB and acquire an immunoregulatory phenotype.

In conclusion, UCB controls glycolysis by dampening the levels of *PGK1* and *ALDOA* glycolytic genes in Th17 differentiated lymphocytes favoring the acquisition of regulatory properties by these cells. In Crohn's disease, silencing of *PGK1* and *ALDOA* restores Th17 cell ability to acquire an immunoregulatory phenotype in the presence of UCB and limits disease activity in a model of colitis in humanized mice. Therefore, blockade of *PGK1* or *ALDOA* should be considered as a potential immunotherapeutic strategy to boost AhR activation and ultimately control Th17 cell inflammatory potential while favoring the acquisition of suppressive features by these cells in Crohn's disease.

## Methods

**Subjects**. Peripheral blood mononuclear cells (PBMCs) and lamina propria mononuclear cells (LPMCs) were isolated from 71 patients with Crohn's disease (45 females and 26 males), recruited from the Gastroenterology Division, Beth Israel Deaconess Medical Center (BIDMC), Boston, MA. Twenty-nine were studied during active disease; the remaining 42 were in remission. At the time of sample collection, 29 patients were on infliximab, one was on adalimumab, six were on steroids, 12 were on mercaptopurine, twelve on ustekinumab, and six were on vedolizumab. Montreal age was scored as "A1" if a diagnosis of Crohn's disease was made before 16 years of age; "A2" if the diagnosis is between 17 to 40 years; and "A3" if the diagnosis was made after 40 years of age. Montreal type was scored as "behavior 1" (B1) in the presence of non-stricturing, non-penetrating disease; and as B2 and B3 in the presence of stricturing and penetrating disease, respectively[46]. Patients' demographic and clinical data are summarized in Supplementary Table 1. PBMCs were also collected from 38 healthy donors (23 females, 15 males, Blood Donor Center at Children's Hospital, Boston, MA). Human studies were approved by the Committee of Clinical Investigations, BIDMC (IRB approval # 2011P000202 ad # 2021P000347). Written informed consent was obtained from all study participants prior to inclusion in the study.

**CD4 cell isolation and T cell polarization**. PBMCs were obtained by density gradient centrifugation on Ficoll-Paque (GE Healthcare Life Sciences, Pittsburgh, PA). LPMCs were isolated from inflamed and non-inflamed colonic areas, obtained from 14 patients with Crohn's disease. The viability of PBMCs and LPMCs was verified by Trypan Blue exclusion and exceeded 98%. Th17 cells were polarized from total CD4 cells, purified from PBMCs and LPMCs by negative selection (Miltenyi Biotec, San Diego, CA). The purity of the immunomagnetically isolated CD4 cells was equal to or higher than 90% in all cases. Purified CD4 cells were resuspended in RPMI1640 (Gibco, Thermo Fisher Scientific, Waltham, MA), supplemented with 10% fetal bovine serum (FBS), and then exposed to Th17 culture conditions for 5 days. These consisted of IL-6 (50 ng/ml), IL-1β (25 ng/ml), TGF-β (3 ng/ml), and Dynabeads Human T activator CD3/CD28 for T cell expansion (Thermo Fisher Scientific) (bead/cell ratio: 1/50). In additional experiments, CD4 cells were polarized into Treg, Th1, and Th2 cells. Polarization into Tregs was obtained upon exposure to IL-2 (100 ng/ml), TGF-β (10 ng/ml), and Dynabeads (bead/cell ratio: 1/2). Polarization into Th1 and Th2 cells was achieved by culturing the cells in the presence of IL-12 (20 ng/ml) and anti-human IL-4 antibodies (10 µg/ml) in the case of Th1 lymphocytes; and IL-4 (10 ng/ml) and anti-human IFNγ antibodies (10 µg/ml) for Th2 cells. Cytokines were all from Peprotech (Rocky Hill, NJ); neutralizing antibodies were purchased from R&D Systems, Minneapolis, MN.

**Cell culture and inhibition studies**. Th17, Tregs, Th1, and Th2 polarized cells were exposed to 20 µM UCB (Frontiers Scientific, Logan, UT) for 6 h, as we have previously shown[5,18]. UCB resuspension and addition to cultures was carried out while limiting exposure to light to the minimum. Cell phenotype, gene expression, and metabolic profile was assessed afterward. In parallel cultures, cells were exposed to glucose at 30 mmol/l[47], and *PGK1* and *ALDOA* mRNA levels were measured afterward. In additional experiments, Th17 cells were exposed to FANA oligonucleotides to selectively silence *PGK1* and *ALDOA* genes. As previously reported[22], FANA oligonucleotides were added at 10 µM. After 72 h incubation at 37 °C and 5% CO$_2$, cells were harvested for assessment of gene expression and phenotype. Other experiments where FANA-*PGK1* and FANA-*ALDOA* were added to CD4 cells on day 2 of their polarization into Th17 cells were performed.

**RNA extraction and NanoString assay**. Total RNA was obtained from peripheral blood derived Th17 cells before and after exposure to UCB, using the RNeasy Mini Kit (Qiagen, Germantown, MD), according to the manufacturer's instructions. RNA quantification was carried out using Nanodrop, whereas RNA integrity was verified using Agilent Bioanalyzer. Samples with DV200 higher than 70% were used in subsequent steps. NanoString profiling was performed using the nCounter Vantage RNA Cancer Metabolism panel (NanoString Technologies, Seattle, WA). One hundred ng total RNA was used as input into hybridization reactions containing reporter and capture probes, as per the manufacturer's recommendations. RNA samples were hybridized at 65 °C for 24 h and subsequently processed and each analyzed in a single nCounter cartridge lane. Post hybridization processing was carried out in a nCounter MAX Analysis System (NanoString Technologies). RCC files were compiled and analyzed using the nSolver analysis software (version 4.0). The internal reference genes included in the probe set were used for normalization. Untreated and UCB treated Th17 cell samples from Crohn's disease patients and controls were run in triplicate and experiments were repeated independently three times.

**qRT-PCR**. Expression of human *ENTPD1/CD39*, *FOXP3*, *PGK1*, *ALDOA*, *IL17A*, *IFNγ*, *IL4*, *IL10*, *PPARα*, *PPARγ*, *AMPK*, and *mTOR* and mouse *Pgk1*, *Aldoa*, *Ifnγ*, *Il17a*, *Tnfα*, and *Il6* was determined by qRT-PCR. Total RNA was obtained from 3–5 × 10$^5$ peripheral blood and lamina propria derived Th17 cells of Crohn's disease patients and healthy controls; and from purified CD4 cells obtained from spleen, MLNs, IELs, and LPs of DSS or vehicle treated C57BL/6 mice, using TRIzol reagent (Thermo Fisher Scientific). mRNA was reverse transcribed using iScript cDNA synthesis kit (Bio-Rad Laboratories, Hercules, CA) according to the manufacturer's instructions. Primer sequences are reported below. *IFNγ* and *FOXP3* primer sequence was as previously indicated[5,48]. The expression of human *IL4* was determined using a TaqMan probe (Thermo Fisher Scientific). Samples were run on a StepOne Plus (Applied Biosystems, Foster City, CA) and results were analyzed by matched software and expressed as relative quantification. Relative gene expression was determined after normalization to human β actin.

Human *ENTPD1/CD39*
Forward 5′ AGGTGCCTATGGCTGGATTAC 3′
Reverse 5′ CCAAAGCTCCAAAGGTTTCCT 3′
Human *PGK1*
Forward 5′ GACCGAATCACCGACCTCTC 3′
Reverse 5′ AGCAGCCTTAATCCTCTGGT 3′
Human *ALDOA*
Forward 5′ CCTACCAATATCCAGCACTGAC 3′
Reverse 5′ CGGTTCTCCTCGGTGTTCT 3′
Human *IL17A*
Forward 5′ AGGCCATAGTGAAGGCAGGAATCA 3′
Reverse 5′ ATTCCAAGGTGAGGTGGATCGGTT 3′
Human *IL10*
Forward 5′ AAGCTGAGAACCAAGACCCAG 3′
Reverse 5′ ATAAGGTTTCTCAAGGGG 3′
Mouse *Pgk1*
Forward 5′ ATGTCGCTTTCCAACAAGCTG 3′
Reverse 5′ GCTCCATTGTCCAAGCAGAAT 3′
Mouse *Aldoa*
Forward 5′ TCAGTGCTGGGTATGGGTG 3′
Reverse 5′ GCTCCTTAGTCCTTTCGCCT 3′

PrimeTime qPCR primers for detecting human *PPARα*, *PPARγ*, *mTOR*, and *AMPK* and mouse *Il17a*, *Ifnγ*, *Tnfα*, and *Il6* were pre-designed by and purchased from Integrated DNA Technologies (Coralville, IA).

**Metabolic assays**. Peripheral blood derived Th17 cells were assayed on an XFe96 Extracellular Flux Analyzer (Seahorse Bioscience, North Billerica, MA) at 4 × 10$^5$/well. OCR and ECAR were determined using the Seahorse XF Cell Mito Stress and XF Glycolysis Stress Kits (Seahorse Bioscience). The XF Cell Mito Stress test included sequential injections of oligomycin (port A), carbonyl cyanide 4-(trifluoromethoxy) phenylhydrazone (port B), and a mix of rotenone and actinomycin A (port C) to inhibit specific complexes of the electron transport chain and provide individual measurements of basal respiration, ATP production, maximal respiration, and non-mitochondrial respiration[49]. The XF

Glycolysis Stress included three sequential injections of D-glucose, oligomycin and 2-Deoxyglucose, as previously reported[50].

**Lactate measurement**. Lactate levels were measured in the culture supernatant of peripheral blood derived Th17 cells before and after exposure to UCB, added to the cells for the last 6 h of culture. Cells were plated at $2 \times 10^5$/well. Supernatants were then collected and stored at $-20\,°C$ until the assay was performed. On the day of the assay, samples were thawed on ice and diluted 50-fold in 1xPBS. The Lactate-Glo Assay (Promega, Madison, WI) was used according to the manufacturer's instructions. Luminescence was read using an Agilent BioTek Synergy 2 multi-mode microplate reader (Agilent, Santa Clara, CA).

**Glucose uptake assay**. Glucose uptake by Th17 cells was determined using the Glucose Uptake-Glo Assay (Promega), as per the manufacturer's instructions. Cells were seeded at $2 \times 10^5$/well and exposed to 20 μM UCB for the last 6 h of culture, as indicated above. After washing with 1xPBS, 1 mM 2-deoxyglucose was added to the cultures. After incubation for 10 min at room temperature, Stop and Neutralization Buffers were added, followed by 2-deoxyglucose-6-phosphate Detection Reagent. Luminescence was recorded after 3 h incubation at room temperature using an Agilent BioTek Synergy 2 multi-mode microplate reader (Agilent).

**Flow cytometry**. The phenotype of differentiated Th17 cells was assessed by flow cytometry. Cells were stained with Brilliant Violet 785 anti-human CD3 (clone # OKT3, Biolegend, San Diego, CA), PE/Cy7 anti-human CD4 (clone # OKT4, Biolegend), PerCP/Cyanine 5.5 anti-human CD4 (clone # A161A1, Biolegend), PE anti-human CCR6 (clone # G034E3, Biolegend), PE anti-human CD39 (clone # A1, Biolegend), and with FITC anti-human IL-23R (cat. # FAB14001F, R&D Systems). Expression of RORC, FOXP3, PGK1, and ALDOA along with the frequency of IL-17A, IL-10, and IFNγ producing lymphocytes among Th17 cells was measured using the eBioscience FOXP3/Transcription Factor Staining Set (Thermo Fisher Scientific), according to the manufacturer's instructions. Following fixation and permeabilization, cells were stained with APC anti-human RORC (clone # AFKJS-9, Thermo Fisher Scientific), APC anti-human FOXP3 (clone # PCH101, Thermo Fisher Scientific), Brilliant Violet 605 anti-human IL-17A (clone # BL168, Biolegend), Brilliant Violet 421 anti-human IL-10 (clone # JES3-9D7, Biolegend), Alexa Fluor 700 anti-human IFNγ (clone # B27, BD Pharmingen), and PE anti-human ALDOA (clone 3F9, Novus Biologicals, Centennial, CO). Staining for human PGK1 was carried out using rabbit anti-PGK1 (clone ST49-07, Novus Biologicals) at 1/50 dilution, according to the manufacturer's instructions. Following incubation for 1 h at room temperature, cells were stained with Alexa Fluor 647 donkey anti-rabbit IgG secondary antibody (clone # Poly4064, Biolegend) at 1/1000 for 30 min.

Cells were acquired on a CytoFLEX LX flow cytometer (Beckman Coulter, Pasadena, CA) and analyzed using FlowJo 2 software (version 10, TreeStar, Ashland, OR). Fluorescence compensation was adjusted based on the fluorescence-minus-one method.

**Induction and assessment of colitis**. Colitis was induced in C57BL/6 mice using DSS. 6-week-old male and female mice were purchased from Taconic (Rensselaer, NY), and housed in our facility up to week 8 or 9. Mice were then exposed to 3% DSS in standard drinking water that was provided ad libitum for 6 days (Fig. 4a). On day 7, DSS was replaced with water for an additional 4 days. Half of the mice were treated with UCB (Frontiers Scientific) at 20 μmol/kg/day intraperitoneally for the whole experiment duration, as we reported previously[5,18,31]. On day 10, mice were sacrificed, colon dissected, and length measured from the ileocecal junction to the anal verge. Spleen, MLN, IELs, and LPs were collected, and CD4 cells were immunomagnetically isolated using the mouse CD4$^+$ T cell isolation Kit (Miltenyi), according to the manufacturer's instructions. The purity of isolated CD4 cells exceeded 90%. CD4 cells were subsequently analyzed for *Pgk1* and *Aldoa* mRNA expression. In preliminary experiments conducted on wild-type mice subjected to vehicle, DSS, or DSS and UCB treatment, expression of *Il6*, *Tnfα*, *Ifnγ*, and *Il17a* was determined by qRT-PCR in CD4 cells isolated from IELs and LPs to confirm the presence of colonic inflammation (Supplementary Fig. 5a–d).

Colitis was also induced using TNBS in *NOD/scid/gamma* immunodeficient mice, preemptively transferred with CD4$^+$ cells that were immunomagnetically isolated from the peripheral blood of one healthy blood donor, as we previously reported[22]. Briefly, 6-week-old female *NOD/scid/gamma* mice were purchased from the Jackson Laboratory (Bar Harbor, ME) and kept under pathogen-free conditions. Mice were injected with $2 \times 10^6$ human CD4 T cells obtained from one healthy blood donor. Prior to injection, cells were verified for expression of *PGK1* and *ALDOA* by qRT-PCR. Three weeks later, mice were bled and checked for human chimerism. Those mice showing more than 10% human chimerism (75% of the originally transferred mice) were sensitized to TNBS (Sigma Aldrich), as we previously reported[22]. One week later, mice were administered a single enema of 0.25 mg TNBS in 50% EtOH (total volume of 50 μl). At the time of TNBS administration, mice were given a single injection of FANA-*PGK1* or FANA-*ALDOA* oligonucleotides at 5.4 mg/kg (or scramble) intraperitoneally[22], UCB at 20 μmol/kg/day or FANA-*PGK1*/FANA-*ALDOA* in combination with UCB (Supplementary Fig. 10a). UCB administration was carried out once a day until the harvest, as per our previous protocols[5,18,31]. Body weight and stool consistency

were checked daily until the harvest, 72 h after the administration of TNBS. In both models, disease activity was calculated based on body weight loss, presence of gross blood, and stool consistency, as we described previously[22]. On the day of the harvest, colons were dissected and length measured. Lymphocytes were collected from the spleen, MLNs, IELs, and LPs for subsequent flow cytometry analysis. Dead cells were excluded using 7-amino-actinomycin D (7-AAD) visibility staining solution (Biolegend) and single-cell gating. Staining was carried out using Brilliant Violet 785 anti-human CD3 (clone # OKT3, Biolegend) and PE anti-human CD39 (clone A1, Biolegend). The expression of FOXP3 was determined using the eBioscience FOXP3/Transcription Factor Staining Buffer Set (eBioscience) and upon staining with APC anti-human FOXP3 antibodies (clone # PCH101, Thermo Fisher Scientific). The frequency of cytokine-producing cells was determined following cell exposure to cell stimulation cocktail plus protein transport inhibitors (500X) (Thermo Fisher Scientific) at 2 μl/ml according to the manufacturer's instructions for 5 h. Staining was then carried out using Brilliant Violet 605 anti-human IL-17A (clone # BL168, Biolegend), Alexa Fluor 700 anti-human IFNγ (clone B27, BD Pharmingen), and Brilliant Violet 421 anti-human IL-10 (clone # JES3-9D7, Biolegend).

Animal protocols were approved by the Animal Care and Use Committee at BIDMC, Boston, MA (protocol # 049-2021).

**Immunohistochemistry staining**. Paraffin-embedded colonic tissue was subjected to antigen retrieval[51]. About 6 μm tissue sections were then stained with hematoxylin and eosin and examined for evidence of colitis. The histology score was calculated as previously reported[5]. Sections were also incubated overnight at 4 °C with polyclonal rabbit anti-human CD3 (cat. # A045229-2, Agilent Dako, Santa Clara, CA) at 1/70. Following endogenous peroxidase blocking with 3% $H_2O_2$, sections were incubated with 1/1000 goat anti-rabbit secondary antibody (Vector Laboratories, Burlingame, CA) for 1 h at room temperature. After treatment with the Vectastain Elite ABC kit (Vector Laboratories), ImmPACT DAB (Vector Laboratories) was applied, and sections were examined by light microscopy.

**Statistics and reproducibility**. For NanoString data, R software (version 4.0.2) was used for the analysis of normalized data, DEG results, and gene set scores obtained from nSolver Advanced Analysis Software. Heatmaps of DEGs (defined based on $P \le 0.05$) were obtained using the Pheatmap package. False discovery rate (FDR) was controlled by the Benjamini–Yekutieli method. All DEGs had FDR lower than 0.05. *PGK1* and *ALDOA* were the genes most significantly impacted by exposure to UCB, based on $P$ value and fold change.

All other results are expressed as mean ± SEM unless otherwise stated. The normality of variable distribution was assessed by Kolmogorov–Smirnov goodness-of-fit test. Comparisons were performed using parametric (paired or unpaired Student's $t$-test) or non-parametric (Mann–Whitney test) tests according to data distribution (all two-sided). One-way ANOVA test, followed by Tukey's multiple comparison test, was used when comparing more than two sets of data. A correlation was made using the Pearson correlation coefficient. $P \le 0.05$ was considered significant. Statistical analysis was performed using GraphPad Prism, version 9.2.0 (GraphPad Software, San Diego, CA).

**Reporting summary**. Further information on research design is available in the Nature Research Reporting Summary linked to this article.

## Data availability

All data that have been generated in this study are included in Supplementary Data 1. For further request, please contact the corresponding Author.

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

## Acknowledgements

Grant support: This work has been supported by the National Institutes of Health (R01 DK108894 and R01 DK124408 to M.S.L.; 1 R21 CA164970 to S.C.R; 1R21TR00175301 to E.K.); Seed Grant Award (Department of Anesthesia, Critical Care & Pain Medicine to M.S.L.); Crohn's and Colitis Foundation (Research Initiative Award to E.K.).

## Author contributions

M.V., N.W., J.J.G., L.G., W.Z. acquisition, analysis, and interpretation of data; drafting of the manuscript; A.K. and L.Z. acquisition and analysis of data; E.C. and J.H. acquisition of data; Y.M., E.K., A.S.C., S.C.R., and M.S.L. critical revision of the manuscript; M.S.L: writing of the manuscript; M.S.L. and S.C.R.: obtained funding.

## Competing interests

The authors declare no competing interests.
