## [Peer Review File · Communications Biology]

Reviewers' comments:

Reviewer #1 (Remarks to the Author):

The paper by Vuerich et al. entitled "Blockade of PGK1 and ALDOA enhances bilirubin control of Th17 cells in Crohn's disease" deals with an important topic of anti-inflammatory and immunometabolic effects of bilirubin. The paper is very innovative and depicts novel mechanisms and cross-talks between immune system and intermediary metabolism, both modulated by bilirubin. Although the paper is well written, important facts related to interpretation of results are not discussed and the paper seems to be over-focused.

Major comments

- 1) Clinical studies on association between IBD and bilirubin concentrations should be discussed as well, such as Zhao et al. *Med Inflamm* 2019; Lenicek et al. *IBD* 2014; Schieffer *PLoSOne* 2017; Tian et al.; *Med Sci Monit* 2018; Su et al. *Medicine* 2019 and De Vries et al. *JCC* 2012, since they provide evidence of clinical impact of observed data by the authors.
- 2) Bilirubin is known agonist of PPAR α /g, PPAR α being a nuclear receptor whose activation has been implicated in the amelioration of experimental IBD (see Basso et al. *Front Immunol* 2021). PPAR α activation was demonstrated to inhibit both PKG1 and ALDOA (as well as other glycolytic genes) (Ribet et al. *Endocrinology* 2010), while PPAR γ was shown to down-regulate PKG1 (see Kumar et al. *J Drug Target* 2011). These data suggest that observed data by authors might be explained also by other mechanisms, which should be discussed in detail. It'd be great if authors could add at least qPCR data on PPAR α /g expressions of the examined immune cells isolated from CD patients, after exposure to UCB.
- 3) Another important metabolic regulator modulated by bilirubin is AMPK, which also plays an important role in pathogenesis of intestinal inflammation (see e.g. Sun and Zhu *Open Biol* 2017). Again, data on AMPK expression in UCB-exposed immune cells would be of big interest.
- 4) Finally, mTOR, an important player in pathogenesis of Crohn's disease (Lashgari et al. *WJG* 2022), is modulated by bilirubin (see Kim et al. *JKMS* 2014). mTOR is known to modulate glycolysis in T cells (Salmond. *Front Cell Dev Biol* 2018). Thus, also this important factor should be discussed and checked as well.

Reviewer #2 (Remarks to the Author):

The authors demonstrated that UCB changed Th17 cell metabolic profiles, especially glycolysis. UCB treatment and silencing PGK1/ ALDOA, the glycolysis-related genes, decreased colitis in the TNBS colitis model. However, there are a number of concerns, which make it unsuitable for publication.

Major concerns:

1. The kit used in Figure 2 was the Mito-stress kit, which is used for measuring the key parameters of oxidation. Therefore, it is not an ideal kit for determining glycolysis. Authors should use the glycolysis kit for measuring different parameters related to glycolysis. The variation of ECAR in CD-TH17 cells was not small (Figure 2A, left lower panel), which might be attributed to no statistical significance. Besides, it is obviously a difference between UCB-TH17 and Untreated-TH17 in CD patients regarding the maximum OCR level (Figure 2A, right lower panel). Authors should increase the sample number and optimize the conditions to minimize the variation in groups. Otherwise, the conclusions that 'decreased in ECAR was found only at later time points' and 'No changes in OCR were noted in CD's derived Th17 after exposure to UCB' are not convincing. The authors stated that the glycolysis-related genes and pathways were increased in CD-TH17 cells compared with HS-Th17 cells (Figure 1). However, it seems like the ECAR and Lactate levels in CD-TH17 cells were similar, even decreased, compared to HS-TH17 cells. The authors need to explain the reasons.
2. The data in Figure 2B only showed that UCB did not change lactate levels in CD-TH17 cells, how do the authors make the indication that 'Th17 unresponsiveness to UCB induced metabolic change'? As

we can see that a total of 31 metabolic genes (6+6+17+2) were changed in CD-TH17 cells after exposure to UCB (Figure 1D). How do the authors define 'Th17 unresponsiveness to UCB'?

3. The staining in Figure 4D, and Supplementary Figures 4 and 5 were not convincing. The authors should show the isotype control and optimize the condition for staining and gating strategy. Considering the bad staining, I cannot trust the conclusion in this part.

4. The data from the last mouse model indicate the potential beneficial effect of silencing PGK1/ALDOA and UCB in colitis, which is a little far from the key story that UCB controls TH17 through glycolysis in CD. The authors might transfer Th17 cells to immune-deficient mice and then test the effect of UCB/or glycolysis on colitis.

Minor concerns:

1. Authors compared the DEGs genes among different group comparisons (HS: Untreated vs UCB; CD: Untreated vs UCB; HS VS CD) in Figure 1D. I would strongly recommend identifying the overlapped DEGs and distinguished DEGs (HS: Untreated vs UCB; CD: Untreated vs UCB;), which helps to reveal the different effects of UCB in T cells from healthy control and CD patients.

2. The efficiency of In-vitro polarization should be included in the manuscript.

3. Does the dose of UCB used in the manuscript affect T cell apoptosis, viability, and proliferation?

4. It would be better to check the glucose uptake levels instead of only checking the expression of Slc2a1.

5. The authors showed that there was a positive correlation between the levels of PGK1 or ALDOA and Montreal type (Figure 3C). Correlated with location? Or other parameters? What's the indication or meaning of these data?

6. Why the value of all the data in the untreated groups was set as 1 in Figures 3D and 3F? It would be more informative if we can see the differences between HS and CD patients.

7. What is the basal glucose level and how much the glucose was added in Figure 3F? Considering the high glucose level might change the cell viability due to the osmotic pressure, the authors should clarify how much glucose they used for the experiments.

8. Please clarify which FANA oligonucleotides were used in Figure 4 and Figure 5? FANA-siPGK1 or FANA-siALDOA? Or the combination?

Reviewer #3 (Remarks to the Author):

In this study, Vuerich et al investigated the role of phosphoglycerate-kinase-1 (PGK1) and aldolase-A (ALDOA) in Unconjugated bilirubin (UCB) control of Th17 cells in Crohn's disease. They showed nicely UCB modulates Th17-cell metabolism by limiting glycolysis and through downregulation of PGK1 and ALDOA. These UCB properties are defective in Th17-cells of CD patients. Silencing of PGK1 or ALDOA restored Th17-cell response to UCB with an increase in FOXP3, IL-10, and CD39. Importantly, the administration of FANA-PGK1 or FANA-ALDOA enhances UCB salutary effects in TNBS colitis in NOD/scid/gamma humanized mice. This is an interesting study, and the data support the conclusion in general. However, some major concerns need to be addressed to improve the quality of the manuscript:

1) Fig1 needs to show the FACS profile of the pushed Th17 cells. Do the pushed Th17 cells have similar levels of IL-17 from healthy control and CD patients, which will greatly affect gene expression profile.

2) Fig2, are the seahorse data from CD or healthy control? Need to show data from both.

3) Fig5D, need to clearly label which are HE and which are CD3 staining.

4) Supplementary Fig2. need to show colonic pro-inflammatory cytokines to confirm the colitis level

5) Supplementary Fig2E. Not clear the indicated gene expressions are in what lymphocytes. Need to show T cell PGK1 or ALDOA expression

6) Supplementary Fig4. IL-17 and Foxp3 staining are a bit odd. Please double-check the staining and gating

7) Supplementary Fig5. Need show dot plots of the FACS profile. The histogram is not ideal.

8) Supplementary Fig6. Please double-check the staining and gating. IL-17 and IL-10 staining is way too high. Foxp3 and IFNg staining is also odd.

Re: COMMSBIO-22-0705-T

Point by point reply to Reviewers

Reviewer # 1

The paper by Vuerich et al. entitled "Blockade of PGK1 and ALDOA enhances bilirubin control of Th17 cells in Crohn's disease" deals with an important topic of anti-inflammatory and immunometabolic effects of bilirubin. The paper is very innovative and depicts novel mechanisms and cross-talks between immune system and intermediary metabolism, both modulated by bilirubin. Although the paper is well written, important facts related to interpretation of results are not discussed and the paper seems to be over-focused.

Author's reply: We thank the Reviewer for the feedback. We have reviewed the comments raised and added new data, as recommended. Please see answer to specific comments below.

Major comments

1) Clinical studies on association between IBD and bilirubin concentrations should be discussed as well, such as Zhao et al. Med Inflamm 2019; Lenicek et al. IBD 2014; Schieffer PLoSOne 2017; Tian et al.; Med Sci Monit 2018; Su et al. Medicine 2019 and De Vries et al. JCC 2012, since they provide evidence of clinical impact of observed data by the authors.

Author's reply: We thank the Reviewer for the comment. We have now added these references, as requested (Introduction, lines 58-63).

2) Bilirubin is known agonist of PPAR α /g, PPAR α being a nuclear receptor whose activation has been implicated in the amelioration of experimental IBD (see Basso et al. Front Immunol 2021). PPAR α activation was demonstrated to inhibit both PKG1 and ALDOA (as well as other glycolytic genes) (Ribet et al. Endocrinology 2010), while PPAR γ was shown to down-regulate PKG1 (see Kumar et al. J Drug Target 2011). These data suggest that observed data by authors might be explained also by other mechanisms, which should be discussed in detail. It'd be great if authors could add at least qPCR data on PPAR α /g expressions of the examined immune cells isolated from CD patients, after exposure to UCB.

Author's reply: We thank the Reviewer for raising this important point. Given the previously reported role of PPAR α and PPAR γ activation in the modulation of glycolytic genes, we have tested the effects of UCB addition on PPAR α and PPAR γ expression in untreated and unconjugated bilirubin (UCB) treated Th17 cells from Crohn's disease patients and healthy controls. No substantial changes in the expression of these genes were noted after Th17 cell exposure to UCB. We have reported these findings in the Results section (lines 155-158), Supplementary Fig. 5a-b, and commented in the discussion (lines 288-291).

3) Another important metabolic regulator modulated by bilirubin is AMPK, which also plays an important role in pathogenesis of intestinal inflammation (see e.g., Sun and Zhu Open Biol 2017). Again, data on AMPK expression in UCB-exposed immune cells would be of big interest.

Author's reply: Thank you for your suggestion. We have now tested the effects of UCB on AMPK levels in Th17 cells of Crohn's disease patients and controls. No significant changes in the expression of AMPK were noted in Th17 cells of both groups after exposure to UCB. We have reported these findings in the Results section (lines 158-161), Supplementary Fig. 5c, and commented in the discussion (lines 291-294).

4) Finally, mTOR, an important player in pathogenesis of Crohn's disease (Lashgari et al. WJG 2022), is modulated by bilirubin (see Kim et al. JKMS 2014). mTOR is known to modulate glycolysis in T cells

(Salmond. Front Cell Dev Biol 2018). Thus, also this important factor should be discussed and checked as well.

Author's reply: Thank you for raising this important point. As recommended by the Reviewer, we have measured mTOR levels in untreated and UCB treated Th17 cells of Crohn's disease patients and controls. No significant changes in the expression of mTOR were noted in Th17 cells of both groups after exposure to UCB. We have reported these findings in the Results section (lines 158-161), Supplementary Fig. 5d, and commented in the discussion (lines 291-294). Please note that instead of quoting the article by Kim et al, in which no obvious change in mTOR levels was observed after bilirubin addition, we have quoted the manuscript by Zhao et al. (Bioorg Med Chem Lett 2021; 51: 128361), who showed decrease in mTOR phosphorylation in the presence of UCB in the cancer setting.

Reviewer 2

The authors demonstrated that UCB changed Th17 cell metabolic profiles, especially glycolysis. UCB treatment and silencing PGK1/ ALDOA, the glycolysis-related genes, decreased colitis in the TNBS colitis model. However, there are a number of concerns, which make it unsuitable for publication.

Author's reply: We thank the Reviewer for the feedback on our work. We have reviewed the comments that have been raised, added new data, and further clarified our findings, as requested. Please see answer to specific comments below.

Major concerns:

1. The kit used in Figure 2 was the Mito-stress kit, which is used for measuring the key parameters of oxidation. Therefore, it is not an ideal kit for determining glycolysis. Authors should use the glycolysis kit for measuring different parameters related to glycolysis.

Author's reply: We have now used the Glycolysis Stress Test to evaluate ECAR of untreated and UCB treated Th17 cells obtained from healthy subjects (n=9) and Crohn's disease patients (n=7). OCR determination was assessed using the Mito Stress Test in the same patients and controls. These new data are reported in the Results section (lines 112-115) and in Fig. 2a.

The variation of ECAR in CD-TH17 cells was not small (Figure 2A, left lower panel), which might be attributed to no statistical significance. Besides, it is obviously a difference between UCB-TH17 and Untreated-TH17 in CD patients regarding the maximum OCR level (Figure 2A, right lower panel). Authors should increase the sample number and optimize the conditions to minimize the variation in groups. Otherwise, the conclusions that 'decreased in ECAR was found only at later time points' and 'No changes in OCR were noted in CD's derived Th17 after exposure to UCB' are not convincing.

Author's reply: We have now increased the sample size for both ECAR and OCR Seahorse determinations and assessed ECAR using the Glycolysis Stress Test, as recommended. These new data are reported in the Results section (lines 112-115) and in Fig. 2a.

The authors stated that the glycolysis-related genes and pathways were increased in CD-TH17 cells compared with HS-Th17 cells (Figure 1). However, it seems like the ECAR and Lactate levels in CD-TH17 cells were similar, even decreased, compared to HS-TH17 cells. The authors need to explain the reasons.

Author's reply: Please note that no statistically significant differences were noted in ECAR, OCR and lactate levels between untreated Th17 cells obtained from patients with Crohn's disease and healthy subjects. These data have now been added to the Results section (lines 113-115). The non-significant decrease in ECAR, OCR and lactate levels in Crohn's derived Th17 cells might be linked to the functional

exhaustion of these cells, a phenomenon that can be often observed in effector cell subsets in chronic inflammatory statuses.

2. The data in Figure 2B only showed that UCB did not change lactate levels in CD-Th17 cells, how do the authors make the indication that ‘Th17 unresponsiveness to UCB induced metabolic change’? As we can see that a total of 31 metabolic genes (6+6+17+2) were changed in CD-TH17 cells after exposure to UCB (Figure 1D). How do the authors definite ‘Th17 unresponsiveness to UCB’?

Author’s reply: This sentence has now been re-phrased and changed with ‘...no differences were noted in the levels of lactate in supernatants of Th17 cells obtained from patients with Crohn’s disease (Fig. 2b), indicating impaired response to UCB in these cells’ (Results section, lines 117-120). Please note that although decrease in the expression of glycolytic genes was noted upon exposure to UCB in Th17 cells obtained from patients with Crohn’s disease, the extent of this decrease was lower than in healthy subject derived Th17 cells (see also Fig. 3d and Supplementary Fig. 3a).

3. The staining in Figure 4D, and Supplementary Figures 4 and 5 were not convincing. The authors should show the isotype control and optimize the condition for staining and gating strategy. Considering the bad staining, I cannot trust the conclusion in this part.

Author’s reply: All data related to flow cytometry analysis have been carefully reviewed and are now presented based on the Reviewer recommendations. The frequency of polarized Th17 cells (Fig. 4d in the original manuscript) is now represented in Supplementary Fig. 1a-b, where information about gating strategy has been provided for both healthy control and Crohn’s derived Th17 cells. Flow cytometry data originally presented in Supplementary Fig. 4 and 5 are now represented in Supplementary Fig. 7 and 8 as pseudocolor plots with fluorescence minus one (FMO) staining as control. Please note that FMO staining has now been provided also for flow cytometry data presented in Supplementary Fig. 4a-b (Supplementary Fig. 1c-d in the original submission) and Supplementary Fig. 10a-g (Supplementary Fig. 7a-g in the original submission).

4. The data from the last mouse model indicate the potential beneficial effect of silencing PGK1/ALDOA and UCB in colitis, which is a little far from the key story that UCB controls TH17 through glycolysis in CD. The authors might transfer Th17 cells to immune-deficient mice and then test the effect of UCB/or glycolysis on colitis.

Author’s reply: We thank the Reviewer for the comments and suggestion. We agree that defining the effects of UCB on Th17 cells in vivo in a T cell transfer model of colitis and determining whether these effects are linked to control of glycolysis or glycolysis-related genes would be important. However, the optimization and establishment of this model (i.e., administration of UCB in a T cell transfer model colitis in Rag immunodeficient mice concomitantly injected with Th17 cells) would not be feasible within the timeframe for revision. We have discussed this important point in the Discussion (lines 279-282).

Minor concerns:

1. Authors compared the DEGs genes among different group comparisons (HS: Untreated vs UCB; CD: Untreated vs UCB; HS VS CD) in Figure 1D. I would strongly recommend identifying the overlapped DEGs and distinguished DEGs (HS: Untreated vs UCB; CD: Untreated vs UCB;), which helps to reveal the different effects of UCB in T cells from healthy control and CD patients.

Author’s reply: Overlapped genes have now been indicated in Fig. 1d.

2. The efficiency of In-vitro polarization should be included in the manuscript.

Author's reply: This information has been added in the Results section (lines 99-101). Flow cytometry plots of polarized Th17 cells obtained from a representative healthy subject and Crohn's disease patient been included in Supplementary Fig. 1a-b.

3. Does the dose of UCB used in the manuscript affect T cell apoptosis, viability, and proliferation?

Author's reply: We thank the Reviewer for giving us the opportunity to clarify this important point. We have previously reported that 20 μ M UCB concentration was chosen as it was associated with less toxicity to Th17 cells, as shown by trypan blue viability test (Longhi et al. JCI Insight 2017; 2: e92791). This UCB concentration decreased Th17 cell proliferation without significantly changing the frequency of apoptotic cells (Longhi et al. JCI Insight 2017; 2: e92791).

4. It would be better to check the glucose uptake levels instead of only checking the expression of Slc2a1.

Author's reply: We have performed the glucose uptake assay in untreated and UCB treated Th17 cells obtained from healthy subjects and Crohn's disease patients. These results are now included in the Results section (lines 122-124) and in Supplementary Fig. 2c and commented in the Discussion (Lines 262-264).

5. The authors showed that there was a positive correlation between the levels of PGK1 or ALDOA and Montreal type (Figure 3C). Correlated with location? Or other parameters? What's the indication or meaning of these data?

Author's reply: We have now revised the information initially provided in the Subjects section and in the Supplementary Table and indicated Montreal type as 'behavior 1' (B1) in the presence of non-stricturing, non-penetrating disease, B2 in the presence of stricturing disease and B3 in the presence of penetrating disease, as per Satsangi et al, Gut 2006; 55: 749-753 (please see Subjects section, lines 363-364 and Supplementary Table 1).

6. Why the value of all the data in the untreated groups was set as 1 in Figures 3D and 3F? It would be more informative if we can the differences between HS and CD patients.

Author's reply: The value of all data in the untreated group in Fig 3d-f was set as '1' to determine the fold change in gene expression of UCB treated vs untreated Th17 cells, as indicated in the corresponding figure legend. We have now presented also the non-normalized data in Supplementary Fig. 3a-d.

7. What is the basal glucose level and how much the glucose was added in Figure 3F? Considering the high glucose level might change the cell viability due to the osmotic pressure, the authors should clarify how much glucose they used for the experiments.

Author's reply: The basal glucose levels in untreated cells was not determined because all the cells were exposed to the same culture conditions. As we have indicated in the Methods section (lines 394-395), cells were exposed to 30 mmol/l glucose, as previously reported (Oleszczak et al, J Recept Signal Transduct Res 2012; 32: 263-270).

8. Please clarify which FANA oligonucleotides were used in Figure 4 and Figure 5? FANA-siPGK1 or FANA-siALDOA? Or the combination?

Author's reply: FANA specifically targeting PGK1 or ALDOA were used in the experiments reported in Fig. 5 (Fig. 4 in the original manuscript) and Fig. 6 (Fig. 5 in the original manuscript). Data obtained in the presence of FANA-PGK1 or FANA-ALDOA are pooled, as indicated in the figure legends of both figures.

Reviewer 3

In this study, Vuerich et al investigated the role of phosphoglycerate-kinase-1 (PGK1) and aldolase-A (ALDOA) in Unconjugated bilirubin (UCB) control of Th17 cells in Crohn's disease. They showed nicely UCB modulates Th17-cell metabolism by limiting glycolysis and through downregulation of PGK1 and ALDOA. These UCB properties are defective in Th17-cells of CD patients. Silencing of PGK1 or ALDOA restored Th17-cell response to UCB with an increase in FOXP3, IL-10, and CD39. Importantly, the administration of FANA-PGK1 or FANA-ALDOA enhances UCB salutary effects in TNBS colitis in NOD/scid/gamma humanized mice. This is an interesting study, and the data support the conclusion in general. However, some major concerns need to be addressed to improve the quality of the manuscript.

Author's reply: We thank the Reviewer for the feedback. We have reviewed the comments raised, added new data, and further clarified our findings, as requested. Please see answer to specific comments below.

1) Fig1 needs to show the FACS profile of the pushed Th17 cells. Do the pushed Th17 cells have similar levels of IL-17 from healthy control and CD patients, which will greatly affect gene expression profile.

Author's reply: We have now indicated the frequency of polarized Th17 cells in the Results section (lines 99-101) and provided flow cytometry plots representing the frequency and phenotype of these cells obtained from healthy subjects and Crohn's disease patients (please see Supplementary Fig. 1a-b). No differences were noted in the frequency and phenotype (i.e., RORC, CCR6 and IL-23R expression levels) of polarized Th17 cells, obtained from healthy subjects and patients with Crohn's disease.

2) Fig2, are the Seahorse data from CD or healthy control? Need to show data from both.

Author's reply: Seahorse data have been obtained for Th17 cell samples from both healthy subjects and Crohn's disease patients, as indicated in Fig. 2a and corresponding figure legend.

3) Fig5D, need to clearly label which are HE and which are CD3 staining.

Author's reply: This information has been provided in Fig. 6d (Fig. 5d in the original manuscript).

4) Supplementary Fig2. need to show colonic pro-inflammatory cytokines to confirm the colitis level.

Author's reply: We have now indicated that in preliminary experiments conducted on wild type mice subjected to DSS colitis or administered vehicle (control), expression of Il6, Tnfa, Ifnγ and Il17a pro-inflammatory cytokines was determined by qRT-PCR in CD4 cells isolated from the intra-epithelial and lamina propria compartments to confirm the presence of colonic inflammation. This information has been added to the Methods section (lines 515-521) and results have been presented in Supplementary Fig. 11a-b.

5) Supplementary Fig2E. Not clear the indicated gene expressions are in what lymphocytes. Need to show T cell PGK1 or ALDOA expression

Author's reply: Pgk1 and Aldoa expression was initially measured in total lymphocytes obtained from spleen, mesenteric lymph nodes, intra-epithelial and lamina propria compartments. As per Reviewer's

request, we now show the expression of *Pgk1* and *Aldoa* in CD4 cells immunomagnetically isolated from these compartments in Fig. 4e. To further emphasize the effects of UCB on *Pgk1* and *Aldoa* also in the DSS mouse model, we now present the data in original Supplementary Fig. 2 as Fig. 4.

6) Supplementary Fig4. IL-17 and Foxp3 staining are a bit odd. Please double-check the staining and gating.

Author's reply: All data related to flow cytometry analysis have been carefully reviewed and are now presented based on the Reviewer recommendations. Flow cytometry data originally presented in Supplementary Fig. 4 are now represented in Supplementary Fig. 7 as pseudocolor plots with fluorescence minus one (FMO) staining as control.

7) Supplementary Fig5. Need show dot plots of the FACS profile. The histogram is not ideal.

Author's reply: Flow cytometry pseudocolor plots are now presented in Supplementary Fig. 8 (Supplementary Fig. 5 in the original submission) with FMO staining as control.

8) Supplementary Fig6. Please double-check the staining and gating. IL-17 and IL-10 staining is way too high. Foxp3 and IFN γ staining is also odd.

Author's reply: All flow cytometry data in the original Supplementary Fig. 6 have been reviewed and are now presented as pseudocolor plots with FMO staining as control (see current Supplementary Fig. 10a-g). We have now indicated that the frequency of IL-17A⁺ (Supplementary Fig. 10a), FOXP3⁺ (Supplementary Fig. 10b), IFN γ ⁺ (Supplementary Fig. 10c), IL-10⁺ (Supplementary Fig. 10d) and CD39⁺ (Supplementary Fig. 10e) cells is measured within the total CD3 compartment (see legend to Fig. 6 and Supplementary Fig. 10). In Supplementary Fig. 10f-g we show the frequency of IL-17A⁺ and FOXP3⁺ cells within the CD3⁺CD39⁺ cell subset (please see also legends to Fig. 6 and Supplementary Fig. 10). Please note that the relatively high frequency of IL-17A⁺ and IL-10⁺ cells might derive from the fact that these are calculated based on the human CD3⁺ cells that are re-populating the mice. Although in all animals used for these experiments human CD3⁺ cells are higher than 10%, the proportion of human lymphocytes is smaller than that originally present in mice. Therefore, the proportion of cells positive for some of the markers appears high because it is measured within human lymphocytes only.

Reviewers' comments:

Reviewer #1 (Remarks to the Author):

The authors addressed all the comments satisfactorily.

Reviewer #2 (Remarks to the Author):

The authors have added some data suggested by the reviewers in this revised paper. However, the data are still not good enough for publication. Below are the specific comments on the revised data.

1. One of the most important concerns is the FACS staining data. The quality of flow cytometry in this paper is suboptimal and thus limits the ability to draw firm conclusions.

a. Supplementary Figure 1: The MFI of IL-17 staining in CD patients was very low, which is different from the staining in healthy controls; RORC is the transcription factor of Th17 cells, and the majority of T cells are not IL-17+ Th17 cells in this figure, but only see one peak of RORC staining, which is not typical staining of RORC.

b. Supplementary Figure 7: The compensation in Supplementary Figure 7a is not good, as we can see some odd shapes. Why is the FSC-A value lower in all the FMO staining compared to other groups? It seems like the gating is not the same in the FMO group.

c. Supplementary Figure 8: Staining is not good.

d. Supplementary Figure 10: Staining is odd, especially Foxp3 and IFN γ .

2. The seahorse data are still not convincing. As shown in Figure 2a, it seems that UCB suppressed OCR in Th17 cells from CD patients, although there was no significant difference. Therefore, I checked the raw data. I noticed that the OCR levels were extremely low, even negative, in several samples of both healthy controls and CD patients, measured at the first time point. Whether the cell number is same among different samples? If the basal level of OCR is too low, the data are not convincing. Not sure if it is a technical issue.

3. The difference in lactate between untreated and UCB groups is too small (See the Raw data, from around 12.1 to 11.8).

4. That UCB has no effect on PPAR α /g, mTOR, and AMPK expression cannot exclude the possibility that these pathways are not involved in UCB induction of glycolysis.

5. In addition to cytokines changes between NO DSS and DSS groups (Supplementary Fig.11), the authors should show the cytokine changes in different treatment groups,

6. There are also several other concerns. For example, I previously asked about the basal glucose level and how much glucose is added to cells shown in Figure 3F. Instead of checking the media they used, the authors just mentioned the same concentration of glucose was used. Therefore, we do not know the final glucose concentration when 30 mmol/l glucose was added. In the paper they cited (Oleszczak et al, J Recept Signal Transduct Res 2012; 32: 263-270), 30 mmol/l glucose was added to the glucose-free media.

Reviewer #3 (Remarks to the Author):

All my previous concerns have been addressed appropriately

Re: COMMSBIO-22-0705A

Point by point reply to Reviewers

Reviewer # 1

The authors addressed all the comments satisfactorily.

Author's reply: We thank the Reviewer for the positive feedback.

Reviewer 2

The authors have added some data suggested by the reviewers in this revised paper. However, the data are still not good enough for publication. Below are the specific comments on the revised data.

Author's reply: Thank you for your feedback. We have reviewed all the comments and amended our manuscript and data presentation accordingly.

1. One of the most important concerns is the FACS staining data. The quality of flow cytometry in this paper is suboptimal and thus limits the ability to draw firm conclusions.
 - a. Supplementary Figure 1: The MFI of IL-17 staining in CD patients was very low, which is different from the staining in healthy controls; RORC is the transcription factor of Th17 cells, and the majority of T cells are not IL-17⁺ Th17 cells in this figure, but only see one peak of RORC staining, which is not typical staining of RORC.

Author's reply: We have now provided a new Supplementary Figure 1, showing the staining of IL-17⁺ cells after CD4 T cell differentiation, in one healthy subject and one patient with Crohn's disease. Please note that the frequency of IL-17⁺ cells after CD4 cell differentiation shown in these representative plots is consistent with previous work (Longhi et al, JCI Insight 2017 and Meng et al, J Immunol 2016). These references have been quoted in the manuscript (lines 99-100). RORC, CCR6 and IL-23R MFI of IL-17⁺ cells is also provided. Details of gating strategy are indicated in the figure legend.

- b. Supplementary Figure 7: The compensation in Supplementary Figure 7a is not good, as we can see some odd shapes. Why is the FSC-A value lower in all the FMO staining compared to other groups? It seems like the gating is not the same in the FMO group.

Author's reply: We have repeated the flow cytometry analysis and chosen new plots to show the frequency of CD39⁺ cells in the IL-17A compartment, and the proportion of FOXP3⁺ and IL-10⁺ cells within the CD3 subset (see new Supplementary Fig. 8).

- c. Supplementary Figure 8: Staining is not good.

Author's reply: We have repeated the flow cytometry analysis and now provide new plots to show the frequency of RORC⁺ and FOXP3⁺ cells within untreated and FANA treated CD4 cells polarized under Th17 conditions (see new Supplementary Fig. 9a-b). Graphs indicating the cumulative data have been also updated (Supplementary Fig. 9a-b).

- d. Supplementary Figure 10: Staining is odd, especially Foxp3 and IFN γ .

Author's reply: Due to time constraints, we could not repeat the colitis model in NOD/scid/gamma immunodeficient mice. However, we have re-analyzed the staining data and provided new flow cytometry

plots to show the frequency of FOXP3⁺ and IFN γ ⁺ cells within the total CD3 cell subset (see new Supplementary Fig. 11b-c).

2. The seahorse data are still not convincing. As shown in Figure 2a, it seems that UCB suppressed OCR in Th17 cells from CD patients, although there was no significant difference. Therefore, I checked the raw data. I noticed that the OCR levels were extremely low, even negative, in several samples of both healthy controls and CD patients, measured at the first time point. Whether the cell number is same among different samples? If the basal level of OCR is too low, the data are not convincing. Not sure if it is a technical issue.

Author's reply: We have repeated the OCR determination in samples of 4 Crohn's patients and 4 healthy subjects, from whom a larger number of PBMCs were available. We confirm our previous findings that addition of UCB significantly decreases OCR in healthy subjects' cells but not in those obtained from patients with Crohn's disease (see updated Fig. 2a and lines 111-112).

3. The difference in lactate between untreated and UCB groups is too small (See the Raw data, from around 12.1 to 11.8).

Author's reply: We have run the lactate assay again in samples obtained from 3 additional healthy controls and 4 additional patients with Crohn's disease. We have now included these data in the graphs presented in the updated Fig. 2b. The data confirm significant decrease in the lactate levels in the culture supernatant of healthy control Th17 cells upon exposure to UCB while no significant changes are noted in Crohn's derived Th17 cells (lines 115-117).

4. That UCB has no effect on PPAR α /g, mTOR, and AMPK expression cannot exclude the possibility that these pathways are not involved in UCB induction of glycolysis.

Author's reply: In our setting, exposure of Th17 cells to UCB does not impact the levels of PPAR α / γ , mTOR and AMPK genes while results in decreased PGK1 and ALDOA levels. We have now clarified in the discussion that although no direct effect of UCB is noted on the expression of these genes, future studies should consider testing whether PPAR α / γ , mTOR and AMPK can modulate glycolysis in a non-AhR dependent manner in Crohn's disease (lines 293-296).

5. In addition to cytokines changes between NO DSS and DSS groups (Supplementary Fig.11), the authors should show the cytokine changes in different treatment groups.

Author's reply: We have now included data showing pro-inflammatory cytokine levels in intra epithelial and lamina propria derived CD4 cells from mice exposed to vehicle, DSS and DSS plus UCB (see new Supplementary Fig. 5).

6. There are also several other concerns. For example, I previously asked about the basal glucose level and how much glucose is added to cells shown in Figure 3F. Instead of checking the media they used, the authors just mentioned the same concentration of glucose was used. Therefore, we do not know the final glucose concentration when 30 mmol/l glucose was added. In the paper they cited (Oleszczak et al, J Recept Signal Transduct Res 2012; 32: 263-270), 30 mmol/l glucose was added to the glucose-free media.

Author's reply: The basal glucose concentration in RPMI medium (Thermo Fisher Scientific, cat. # 11-875-085) is 11.11 mmol/l. In our experiment, we added 30 mmol/l glucose, as indicated in the Methods (lines 396-397).

Reviewer # 3

All my previous concerns have been addressed appropriately.

Author's reply: We thank the Reviewer for the positive feedback.

REVIEWERS' COMMENTS:

Reviewer #2 (Remarks to the Author):

Most of the concerns were addressed by the authors.

Reviewer #3 (Remarks to the Author):

All my previous concerns have been addressed appropriately

Re: COMMSBIO-22-0705B

Point by point reply to Reviewers

Reviewer 2

Most of the concerns were addressed by the authors.

Author's reply: We thank the Reviewer for the positive feedback.

Reviewer # 3

All my previous concerns have been addressed appropriately.

Author's reply: We thank the Reviewer for the positive feedback.